# Controlled Reservoir Drawdown—Challenges for Sediment Management and Integrative Monitoring: An Austrian Case Study—Part A: Reach Scale

**Christoph Hauer [1],\*, Marlene Haimann [1], Patrick Holzapfel [1], Peter Flödl [1] 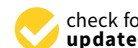, Beatrice Wagner [1], Michael Hubmann [2], Bernhard Hofer [3], Helmut Habersack [1] and Martin Schletterer [3,4]**

[1] Christian Doppler Laboratory for Sediment Research and Management, Institute of Hydraulic Engineering and River Research, Department of Water-Atmosphere-Environment, University of Natural Resources and Life Sciences, Vienna (BOKU), Muthgasse 107, 1190 Vienna, Austria; marlene.haimann@boku.ac.at (M.H.); patrick.holzapfel@boku.ac.at (P.H.); peter.floedl@boku.ac.at (P.F.); beatrice.wagner@boku.ac.at (B.W.); helmut.habersack@boku.ac.at (H.H.)

[2] H&S Limnologie GmbH, Valiergasse 60, 6020 Innsbruck, Austria; m.hubmann@limnologie.at

[3] TIWAG-Tiroler Wasserkraft AG, Eduard-Wallnöfer-Platz 2, 6020 Innsbruck, Austria; bernhard.hofer@tiwag.at (B.H.); martin.schletterer@tiwag.at (M.S.)

[4] Institute of Hydrobiology and Aquatic Ecosystem Management, University of Natural Resources and Life Sciences, Vienna (BOKU), Gregor-Mendel-Straße 33, 1180 Vienna, Austria

\* Correspondence: christoph.hauer@boku.ac.at; Tel.: +43-1-47654-81912

**Abstract:** For Europe, a reduction of 80% of the potential storage volume due to reservoir sedimentation is predicted by 2080. Sedimentation processes trigger the decrease of the storage volume and a related restriction in hydropower production. Further, the artificial downstream flushing of deposited fines has manifold effects on the aquatic ecology, including changes in morphology and sediment quality, as well as increased turbidity and subsequent stress for aquatic species. However, it is common to lower the water surface of reservoirs for technical inspections, which is not comparable to reservoir flushing operations. The presented case study deals with such a controlled drawdown beyond the operational level of the Gepatsch reservoir (Tyrol, Austria). Based on the awareness of possible ecological consequences, an advanced set of measures and an integrative monitoring design, consisting of a detailed event-based quantification of suspended sediments, changes in the morphology, especially with respect to fine sediments, and analyses of the biological quality element fish on the reach scale along the Inn River have been developed.

**Keywords:** reservoir management; hydropower; flushing vs. controlled drawdown; fine sediment dynamics; aquatic biota

## 1. Introduction

The sedimentation of reservoirs causes diverse problems from an operational perspective. Thus, this is one of the most significant global issues concerning the future use and development of hydropower [1] as well as in water engineering in general. A few decades ago, the basic question of the lifetime of artificial reservoirs was hardly addressed by sedimentologists and hydraulic engineers [2]. As result of this omission, the sedimentation of reservoirs has become a global challenge. This is especially true for Asia, a continent with high erosion rates, where a reduction of 80% of the potential storage volume by 2035 has been predicted. The same is forecast for Europe by 2080 [3]. The lifetime of reservoirs is characterized by [2] into four classes ranging from (i) very short (10–20 y), e.g., [4]; (ii) short (20–60 y), e.g., [5]; to (iii) medium (60–200 y), e.g., [6]; and (iv) long (>200 y), e.g., [7]. However,

climate change and the subsequent retreat of glaciers in alpine regions are additional drivers that may influence the sediment supply rates, as additional sediment inputs have occurred from recent glacial deposits [8].

In addition to a reduction of the storage volume [9] and the related effects on hydropower production [10], the deposition of sediment in reservoirs supports a lack of substrate diversity downstream [1]. Sustainable sediment management in reservoirs and regulated rivers were summarized by Kondolf et al. [11]. In their review, the authors listed measurements related to (i) bypassing sediment around the reservoir and (ii) sluicing (or drawdown routing) which permits sediment to be transported through the reservoir rapidly to avoid sedimentation during high flows. They also presented (iii) drawdown flushing, which involves scouring and re-suspending sediment deposited in the reservoir, and transporting the sediments downstream through low-level gates in the dam and (iv) turbidity currents vented through the dam, as sustainable sediment management measures. The downstream flushing, however, of deposited fines can negatively affects the aquatic ecology [12], although, (regular) sluicing diverts the floods and sediments directly into the receiving water course. Thus, from an ecological point of view, sluicing is one of the preferable options. Many studies have focused on sedimentological changes [13] or investigated the impact on the biota in detail, e.g., [14]. The flushing of sediment is one opportunity for so-called active management by the establishment of (i) density current venting [15], (ii) configuration of reservoirs [16], and (iii) sediment bypass tunnels [17,18]. However, here not only fines are diverted but also bedload [19]. The ecological consequences of an increased surplus of fines were studied in laboratories, e.g., [20,21] as well as in field investigations, e.g., [22,23]. Some studies showed that recovery might be high after a short time. Hence, long-term (regular) sediment management is beneficial also for benthic biota [24–27].

The direct impacts of flushing on biota can be related to various trophic levels, such as changes in algal and macrophyte communities on the lower trophic levels, which might result from human interventions via a build-up of an excessive fine sediment supply [28,29]. On a higher level, macroinvertebrates can be impacted due to smothering of the substrate and clogging of the subsurface layer, denoted as the interstitial layer [30–32]. Also, on a higher trophic level, fish might be affected by reduced feeding activity due to turbidity [33,34], by reduced reproduction success due to fines on spawning grounds [35], or by mechanical damage of their gills [22]. The general focus of sediment management in reservoirs is mostly given on fines (diameter < 0.5 mm), which are transported as a suspended load in terms of remobilization [11,36]. Thus, the impact assessment frequently addresses the suspended sediment concentration (SSC), which defines the total amount of both the mineral and organic material carried in suspension by an open channel flow [37]. However, independent of the reservoir impacts, the transport of suspended sediments in a river environment is highly non-linear in time and space, e.g., [38] and is influenced by many factors, including the geological framework, climatic conditions, and the topography of the drainage basin [39,40]. In addition to the well-known single-cause relationships for various aspects in sediment management of reservoirs, only minor studies, e.g., [41] deal with a detailed integrative assessment of sediment management measures. An integrative view of the downstream impact on the abiotic and biotic environments (on various trophic levels) was only conducted in one comparable study, the controlled drawdown in winter of the Cancano reservoir, Italy [42]. Moreover, almost no recommendations for an appropriate monitoring design are available in the literature.

It is important to differentiate between (i) reservoir flushing and (ii) a controlled drawdown of a reservoir, but a common terminology is lacking. While reservoir flushing has the aim to remove the sediments [43–45], which takes place at high flow rates for short periods of time (i.e., suspended sediment concentrations are high), a controlled drawdown of a reservoir is usually conducted during a period of low flow (i.e., suspended sediment concentrations are relatively low, but (depending on the reservoir size) the time of the emission is longer [46]. From an operational point of view, a controlled drawdown is only possible during reduced inflow rates, as high inflows would result in high mobilization of fines. For alpine regions, this means only during winter, where snowfall at high

altitudes limits the catchment run-off, and thus the discharges into the reservoir [47]. However, winter is a very sensitive period for aquatic organisms, e.g., [48], hence flushing during these cold months is not targeted [49]. This is another novel aspect of the present study, where a focus was set on the question of how aquatic organisms react to increased SSCs during a controlled drawdown in winter (low-flow period).

The presented case study deals with a controlled drawdown of the reservoir Gepatsch (Tyrol, Austria) beyond the operational level. In this context, an increased load of suspended sediments—with ecological impacts on the downstream biota as well as technical consequences related to turbine abrasion during the drawdown—was expected. The research questions, which are addressed in the presented study are: how does the possible increase in fine sediment concentrations affects both the hydro-morphological environment and biota on the reach scale. Based on the awareness of possible impacts, an advanced set of measures and an integrative monitoring concept was developed, both for ecological and technical aspects.

## 2. Materials and Methods

### 2.1. Case Study

The hydropower plant (HPP) Kaunertal is a high-pressure storage power plant for the annual storage reservoir Gepatsch (storage volume approx. 140 million m$^3$), located in the Kaunertal valley, Tyrol (Austria) [50]. The natural catchment area of the Gepatsch reservoir is 107 km$^2$ and due to a diversion system (intakes from neighboring catchments), the catchment area was increased to 279 km$^2$ [51]. The Kaunertal HPP was built from 1961 to 1965 as a single-stage plant with an approximately 900 m head between the Gepatsch reservoir and the Prutz power station (capacity = 392 MW). It generates peakload energy [52] and with an average annual production of 661 GWh, the Kaunertal HPP is still one of the most powerful installations in Austria. The last time the upper and lower intakes and the bottom outlet were accessible, was during a reservoir flushing in 1977. The complete drawdown of 2015/2016 was stipulated by authorities in an official inspection ruling, with the objectives of checking the components of the dam that cannot be seen during normal operation, checking the reservoir slopes and the structural operation equipment, and carrying out maintenance measures [46]. Due to the occurring inflow conditions, the complete drawdown was only possible in winter (approximately 0.5 m$^3$ s$^{-1}$ in the winter months). In summer this could not have been performed because of the up to 50 times higher inflows as a result of glacier melting. The water level was lowered beyond the operational level as a controlled drawdown via the intakes, not via the bottom outlets, in order to minimize negative effects on the river Fagge. The retained water was convoyed via the plant's pressure tunnel to Prutz into the largest receiving water course, the Inn River. However, as this procedure might have been harmful to the turbines, below a certain reservoir level the water was discharged only towards one of the five turbines to restrict abrasion damage to a single turbine. In this regard, an intensive monitoring of turbine abrasion was performed to identify the influence of the transported sediments on the turbine wear [53,54].

The so-called "free reservoir management" applies to the Gepatsch reservoir from a full supply level (1767 m a.s.l.) down to a water level elevation of 1665 m a.s.l., i.e., operation of the reservoir between storage level and minimal operating level. As part of the controlled drawdown, detailed monitoring and management began at a water level height of 1690 m a.s.l., which was reached on 7 December, 2015. The core of the real-time monitoring was the measurement of the discharge of the suspended sediment load into the Inn River. Thresholds for the reservoir drawdown between December 2015 and March 2016 were stipulated by the authorities. The target value for the permanent SSC immission was 1 g L$^{-1}$ (as the drawdown took place in the spawning period of the brown trout), but up to 10 g L$^{-1}$ was permissible in the short term, i.e., a maximum of 3 g L$^{-1}$ for a period of up to 24 h, a maximum of 5 g L$^{-1}$ for a period up to 6 h, and a maximum of 10 g L$^{-1}$ for a period up to 2 h. Steps to reduce high SSCs were taken immediately when the guidance value was exceeded

for the permanent immission. As the input into the Inn River was stopped at weekends, "recovery phases" were provided for the aquatic biota [46], which is also known from other case studies [55]. Further thresholds were set by the authorities for the dissolved oxygen concentration (permanent value above 10 mg L$^{-1}$ (maximum 2 h between 10 and 6 mg L$^{-1}$), for ammonium (permanent value below 0.3 mg L$^{-1}$, up to 0.5 mg L$^{-1}$ for a maximum of 2 h) and for pH (allowed range between 6 and 9). Within the herein presented case study, a large stretch of the Tyrolean part of the Inn River was investigated in order to assess the longitudinal effects (Figure 1). Over the last few centuries multiple hydro-morphological pressures, such as land reclamation, flood protection, establishment of infrastructure (e.g., railways and highways) as well as hydropower development have taken place along the Inn River similar to other rivers in the Alps (compare to [55,56]). Hence, the river course in the province of Tyrol is classified as a heavily-modified water body [57].

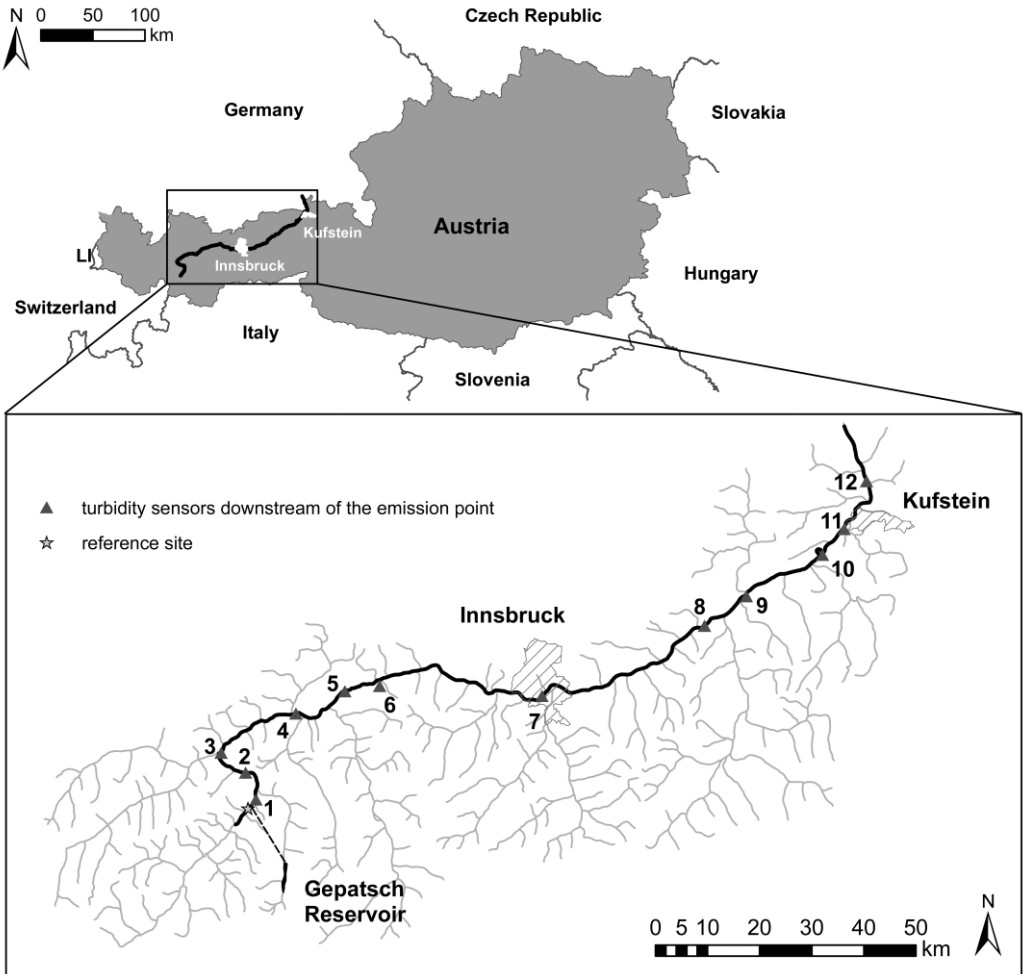

**Figure 1.** Austrian map with the sampled Inn River highlighted and reach scale sampling stretch of the Inn River from the reference site Ried (upstream boundary) to the German border (downstream boundary). The penstock from the reservoir to the power station in Prutz is dotted (the monitoring station in the tailrace channel is not indicated). The triangles indicate continuous turbidity monitoring stations (at gauging stations). 1 = Prutz Entbruck, 2 = Fließ Neuer Zoll, 3 = Landeck−Perjen, 4 = Karrösten Königskapelle, 5 = Haiming Magerbach, 6 = tailrace channel HPP Silz, 7 = Innsbruck Arthur Haidl Promenade, 8 = Rotholz, 9 = Rattenberg, 10 = Kirchbichl, 11 = Langkampfen, and 12 = Oberaudorf.

## 2.2. Monitoring Concept

The monitoring during the controlled drawdown of the Gepatsch reservoir was established based on the two defined research questions and in order to fulfil the governmental requests (i.e., clauses related to the permit) (see Figure 2). The monitoring was set up into (i) a reach scale-monitoring network for continuous turbidity measurement, fine sediment deposits, and tributary connectivity (presented herein) and (ii) local-scale monitoring of cross-sectional variability as well as habitat-related turbidity and sedimentological/ecological analyses in general [58], conducted as pre- and post-event monitoring.

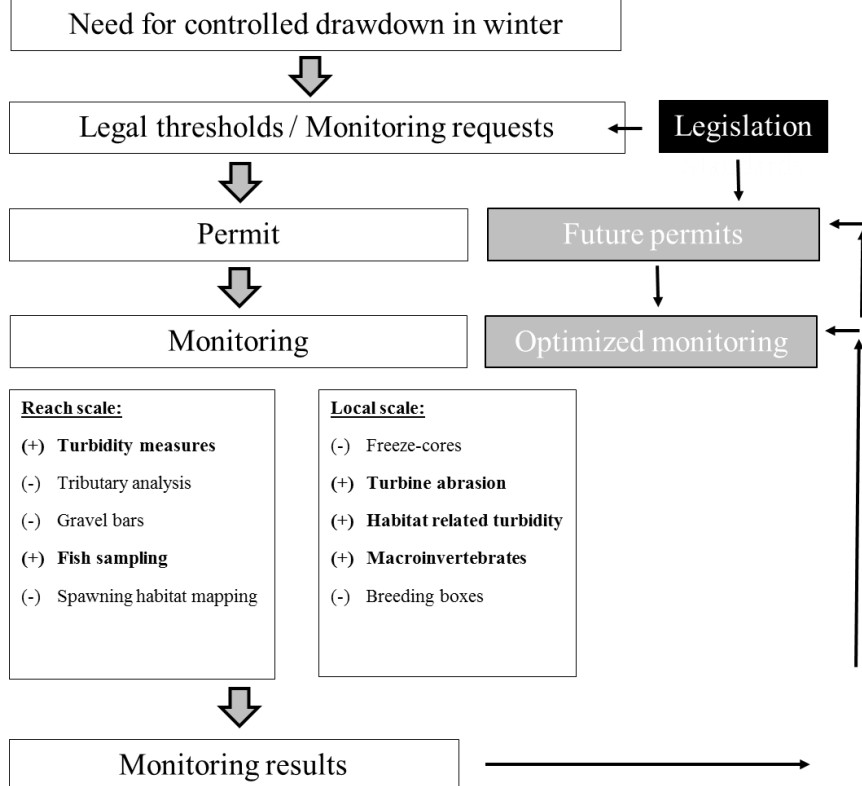

**Figure 2.** Flow chart for the controlled drawdown monitoring concept (CDMC) of the Gepatsch reservoir (white boxes) and the drawback of the conducted research for future permits and definition of optimized monitoring strategies (gray boxes). The role of governmental actions is presented in the black box. (+) indicates factors that are important for the ecological impact assessment in terms of a controlled reservoir drawdown and (−) indicates factors that are of minor importance for the ecological impact assessment in terms of a controlled reservoir drawdown.

## 2.3. Reach Scale

### 2.3.1. Continuous Monitoring of SSCs: Turbidity Measurements

The turbidity measurements were conducted with turbidity sensors (Solitax ts-line sc from Hach Lange GmbH, Düsseldorf, Germany) at twelve gauging stations along the study reach (Figure 1). This sampling device measures both the turbidity in FNU (Formazine Nephelometric Units) and the SSC (g L$^{-1}$) using the 90° scattered light method or the color-independent duo scattered light measuring method. The measuring range of this device is from 0.001 to 4000 FNU or 0.001 to 50 g TS L$^{-1}$. Without calibration (zero adjustment), a measurement accuracy of <5% of the measured value can be achieved. With calibration an accuracy of <1% of the measured value can be achieved (see [57,59,60]). The measured values were transformed into an analogue signal (4–20 m a.s.l. and transmitted by means of network control technology (Siemens SAT AK 1703). From the 1-minute values (mean value

based on continuous measurements) a 15 minute-average was calculated. If at least 80% of the minute values were present, the resulting mean value was calculated, which was subsequently passed on.

A disadvantage of the turbidity sensors is that small grain sizes cause a relatively high turbidity value at a relatively low mass, while coarser grains cause relatively low turbidity due to the smaller specific surface area [61]. The occurrence of organic matter has an additional influence on the measured turbidity by the instrument. The turbidity line must therefore be converted to a concentration curve using samples obtained near the probe (calibration samples). In the case of direct sampling (single-point sampling) a suspended sediment sample was taken at one point in the river using a point-integrating or an instantaneous-sampling device for calibration of the turbidity sensors concerning functional relationships between the turbidity and SSC. An important fact regarding sampling is to obtain isokinetic correct samples to avoid over- or under-sampling of the probes, e.g., [62,63]. The single-point sampling near the sediment sensors took place from the banks. To record the temporal variability and to calibrate the turbidity sampling recordings, repeated measurements were conducted (n > 30 for each station) for different discharge rates. The measuring network, which was installed at the Inn River during the drawdown of the Gepatsch reservoir and was used for investigations, is shown in Figure 1. The monitoring network was established on the basis of different legal requirements (e.g., Water Framework Directive, Austrian Water Act), as well as additional provisions (e.g., fishing rights), resulting in broad scientific data sampling. During the drawdown two monitoring locations were important for the operation of the intakes, one at the reference site at Ried (see Figure 1) and another one in the tailrace channel. Based on the SSCs and discharge rates from those stations, the SSCs at the gauging station Prutz were calculated in real time (Equation (1)). At the gauging station Prutz the threshold criteria of the permit (see Section 2.1) were controlled in real-time and a weekly documentation was compiled.

$$\text{SSC}_{\text{PRUTZ}} =$$
$$[(Q_{\text{RIED}} \times \text{SSC}_{\text{RIED}}) + (Q_{\text{TAILRACE HPP PRUTZ}} \times \text{SSC}_{\text{TAILRACE HPP PRUTZ}})]/(Q_{\text{RIED}} + Q_{\text{TAILRACE HPP PRUTZ}}) \tag{1}$$

where $\text{SSC}_{\text{PRUTZ}}$ is the calculated suspended sediment concentration (SSC) at the Prutz gauging station, $\text{SSC}_{\text{RIED}}$ and $\text{SSC}_{\text{TAILRACE HPP PRUTZ}}$ are the measured suspended sediment concentrations at the Ried station and at the tailrace channel of the HPP Kaunertal, respectively. $Q_{\text{TAILRACE HPP PRUTZ}}$ is the measured discharge rate, while $Q_{\text{RIED}}$ is calculated ($Q_{\text{RIED}} = Q_{\text{PRUTZ}} - Q_{\text{HPP PRUTZ}} - Q_{\text{COOLING WATER}}$). $Q_{\text{TAILRACE HPP PRUTZ}}$ consists of the discharge from the hydropower plant ($Q_{\text{HPP PRUTZ}}$) and the discharge from the cooling water ($Q_{\text{COOLING WATER}}$).

Accordingly, the amount of discharge from the Gepatsch reservoir was regulated by Equation (2b), based on the following considerations:

$$\text{SSC}_{\text{TARGET [1000 mg/l]}} \times Q_{\text{PRUTZ}} =$$
$$\text{SSC}_{\text{TAILRACE HPP PRUTZ}} \times (Q_{\text{HPP PRUTZ}} + Q_{\text{COOLING WATER}}) + \tag{2a}$$
$$\text{SSC}_{\text{RIED}} \times (Q_{\text{PRUTZ}} - Q_{\text{HPP PRUTZ}} - Q_{\text{COOLING WATER}})$$

Thus,

$$Q_{\text{HPP PRUTZ}} =$$
$$[Q_{\text{PRUTZ}} \times (\text{SSC}_{\text{TARGET [1000 mg/l]}} - \text{SSC}_{\text{RIED}}) + Q_{\text{COOLING WATER}} \times (\text{SSC}_{\text{TARGET [1000 mg/l]}} - \text{SSC}_{\text{RIED}})] / \tag{2b}$$
$$[\text{SSC}_{\text{TAILRACE HPP PRUTZ}} - \text{SSC}_{\text{Ried}}]$$

where the target value for the permanent SSC immission ($\text{SSC}_{\text{TARGET [1000 mg/l]}}$) is also included, which enabled to change the flow rate from the Gepatsch reservoir ($Q_{\text{HPP PRUTZ}}$) depending on the SSC target concentrations in real-time.

In addition to the SSC measurements, the gauging station Prutz (see Figure 1) also continuously monitored the physicochemical parameters ammonium (WTW, AmmoLyt®Plus 700 IQ), oxygen content and oxygen saturation (WTW, FDO®700 IQ) as well as pH (WTW, SensoLyt®700 IQ); all sensors were manufactured by Xylem Analytics Germany Sales GmbH & Co. KG, WTW, Weilheim,

Germany. All of these sensors were fixed with steel chains at the bridge in order to be operated at different water levels.

### 2.3.2. Fine Sediment Deposits on Gravel Bars

In November 2015 (pre-monitoring) and March 2016 (post-monitoring) field investigations were conducted along the Inn River between kilometers 386 and 299 (87 km out of the total 160 km of the monitoring reach) (Figure 1). In total 36 gravel bars were selected. At each gravel bar a minimum of n = 30 measurements of vertical height of fine sediment deposits were taken in a random walk approach; each sampling point was GPS-tracked and visualized in the GIS-systems. To determine the height of the fine sediment deposits, a scaled iron ruler (diameter = 1 cm) was driven mechanically into the fines until it stopped at the coarse surface layer of the gravel bar. Random measuring points on each gravel bar were pooled for each gravel bar site (n = 36) and subsequently statistically analyzed. The statistical parameters (max, min, mean and standard deviation) were calculated and boxplots were created in order to compare the results. For the pre- and post-event periods, the testing procedure of a Mann–Whitney U test for two independent samples was employed after checking for normal distributions using a Shapiro–Wilk normality test [64] and examining the homogeneity of variances using Levene's test [65]. A non-parametric test was conducted in order to test the null hypothesis that the height of the fine sediment layer varied significantly ($p < 0.05$), via, e.g., increased deposits after the emptying of the reservoir.

### 2.3.3. Mapping and Analysis at Salmonid Spawning Sites

Potential gravel spawning places were mapped and exemplarily sampled by volumetric grab-sampling using a USGS standard format of 60 cm × 60 cm sampling frame as part of the pre-event monitoring in November 2015. The investigation area extended from the Kaunertal tailrace ~ 90 km downstream (city of Innsbruck) (Figure 1. Potential gravel spawning sites were marked by GPS points (Garmin Oregon 600), and special properties, such as the size of the spawning site, were documented in a database. At suitable points along the Inn River (Prutz, Fließ, Karrösten, and Stams) as well as the tributary Fagge (as a reference), sampling was carried out. Sampling focused on obtaining grain size distributions of the surface- and subsurface layer at representative spawning sites (n = 5). The samples were dried and sieved in order to compare the grain size distribution to the international documented characteristics of the salmonid spawning sites; see e.g., [66].

### 2.3.4. Fish—Abundance and Biomass

The quantitative fish surveys on the reach scale were conducted using a Water Framework Directive compliant methodology [67]. At most sites (Figure 1 Table 1), electrofishing was performed by boat (strip-fishing method). Due to the dimensions of the river, at most sites a boat was needed, and strips were fished (fishing gear: 7-kW-unit with 300/600 V direct current) according to [68], which was accompanied by a wading team on the banks (fishing gear: 1.5-kW-unit with 300/600 V direct current). Based on the criteria density (biomass, kg ha$^{-1}$), species (expected vs. observed species), dominance (based on the fish region index), and population structure (for each species) the "Fish Index Austria" (FIA) [67] was calculated for the pre- and post-monitoring data.

**Table 1.** Description of the various investigated sites including fish region, national classification, river length, and date of the pre- and post-monitoring.

| Site | Fish Region (According to BAW 2010 [69]) | Category (According to Haunschmid et al. 2015 [67]) | Section (rkm) | Pre-Monitoring | Post-Monitoring |
|---|---|---|---|---|---|
| Ried | Hyporhithral big | C | 388.76–409.74 | 07.12.2015 | 28.11.2016 |
| Prutz | Hyporhithral big | C | 382.28–288.76 | 20.11.2015 | 18.11.2016 |
| Fließ | Metarhithral | A3 (2015) B4 (2016) | 378.31–382.28 | 10.11.2015 | 15.11.2016 |
| Landeck | Hyporhithral big | C | 357.25–374.27 | 09.11.2015 | 26.11.2016 |
| Imst | Hyporhithral big | C | 334.29–357.25 | 29.11.2015 | 27.11.2016 |
| Silz | Hyporhithral big | C | 334.29–357.25 | 15.11.2015 | 13.11.2016 |
| Mils / Hall | Hyporhithral big | C | 256.01–294.59 | 23.11.2015 | 21.11.2016 |
| Brixlegg (downstream Ziller) | Epipotamal | C | 236.14–256.01 | 16.11.2015 | 07.11.2016 |
| Langkampfen / Kufstein | Epipotamal | C | 217.60–236.14 | 14.12.2015 | 02.11.2016 |

### 2.3.5. Tributary Analysis

In total 53 selected tributaries were assessed for their hydro-morphological quality at their mouth into the Inn in order to investigate their potential as refugial sites. The investigations were performed by GIS-based mapping in the field. The goal was to evaluate the theoretical migration possibility (during the winter's low-flow period) for fish from the Inn River, which could be used during the drawdown. The data basis for the tributaries was the dataset of the state of Tyrol, which includes water bodies and catchment areas in the province of Tyrol at a scale of 1:10,000 [70]. With the open-source geographic information software QGIS and a pre-selection of n = 180 tributaries to the Inn River was made. In the second step, tributaries with a catchment area of less than 10 km$^2$ were excluded from the analyses. This resulted in a selection of n = 53 tributaries in the study reach. During fieldwork, only the mouth and an upstream distance of 50 m of the tributaries were assessed. The following parameters were mapped: (i) connectivity, (ii) upstream hydro-morphological characteristics (substrate and morphological diversity), and (iii) the occurrence of salmonid spawning gravel (grain diameter, 2–5 cm) on suitable spawning sites; see [71].

## 3. Results

### 3.1. Suspended Sediment Concentrations and Sediment Transport

The controlled drawdown can be divided into two characteristic periods: (i) the drawdown itself from 07 December, 2015 to 13 January, 2016 and (ii) a cleaning phase (removal of fine sediments from the penstock, etc.) from 14.01.2016 to 14.03.2016. During the drawdown large volumes of fine material and silt were transported towards the intake structures and filled the dead storage capacity with about 400,000 m$^3$ of sediment, thus the controlled drawdown had to be stopped to guarantee the operational safety [72]. Figure 3a shows the suspended sediment concentrations for the gauging station Prutz and the related values from the permit (see Section 2.1.; case study). In addition, Figure 3a shows the periods of the "recovery phases", i.e., no discharge into the Inn River, on weekends (gray) as well as due to cleaning of the waterway (yellow). Related to the opening of the valve after the weekends, higher values appeared due to sedimentation processes in the penstock as result of the shutdown (closing the valves). This indicated that it would have been better not to stop the flow completely but simply reduce it during the weekends.

The graphic shows, that during the drawdown the SSCs immission at the gauging station Prutz was within the target value (i.e., below 1 g L$^{-1}$) with some short peaks ($\frac{1}{4}$ h mean values) due to both drawdown and cleaning. In case values above 1 g L$^{-1}$ were observed (i.e., single 15 minute mean-values), immediately the discharge was reduced. Also sometimes, due to the calculation method, the higher values could have been wrong. As the stretch is affected by hydropeaking it could have been

the case that between 00:00 and 00:10 a base flow of 5 m$^3$/s occurred, while from 00:10 till 00:15 a peak flow with 90 m$^3$/s occurred. Thus the discharge from the drawdown was low in the time window 00:00 and 00:10, but as the high discharge was available in the receiving water course the discharge from Gepatsch reservoir could also be increased. However, sometimes this resulted in "wrong" (i.e., too high) calculated $\frac{1}{4}$ h mean values. This phenomenon was observed sometimes and could be validated with measured data from the gauging station Prutz.

Overall, the values were below the SSCs thresholds, which had been determined in the official permit. However, not only was the legal perspective of the recorded data of importance, but even more important was how the recorded data fitted to the natural suspended sediment dynamics of the Inn River. Hence, in Figure 3b, the mid-term SSC of the Inn River at the monitoring station Innsbruck is highlighted. This station was chosen because it has the longest SSC time series in Tyrol, i.e., with ten years of monitoring data (2005–2015), and clearly exhibits low suspended sediment concentration rates during winter, but also in winter naturally high values of 1600 mgL$^{-1}$ can occur. Thus, the controlled drawdown overtopped the statistical mean values of the winter months. However, the duration of overtopping (SSC immission during the controlled drawdown) with a target value of 1 g L$^{-1}$ was within the natural range and did not harm biota (see Section 3.4 and Part B [58]).

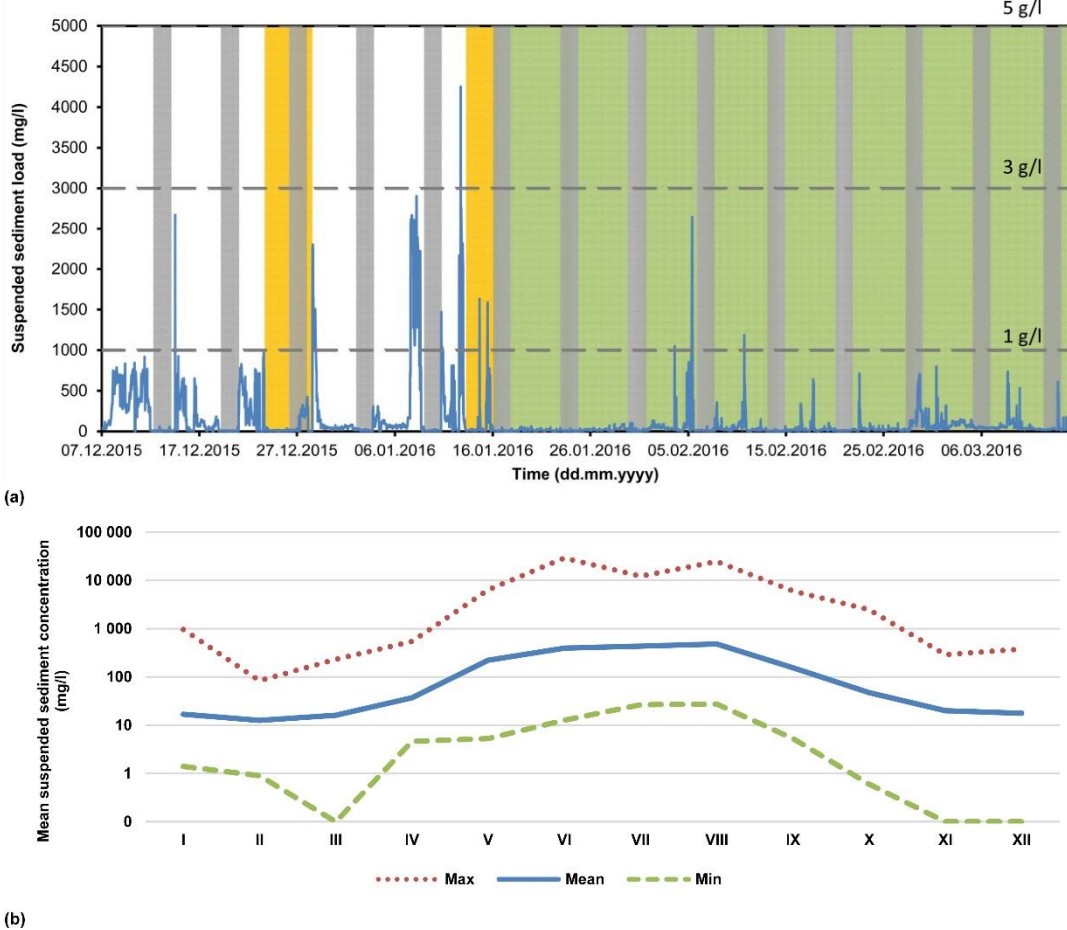

**Figure 3.** (**a**) Continuous records of suspended sediment concentration at sampling station 1 (Prutz) (immediately downstream of the tailrace channel of the HPP Kaunertal) (blue line); yellow bars = stop of the continuous drawdown via the penstock and; green bars = cleaning period and (**b**) minimum, mean, and maximum suspended sediment concentration per month at gauging station Innsbruck for monitoring period 2005–2015 (database: hydrological survey Tyrol); y-axis: log-scale.

Besides the data at the gauging station Prutz, immediately downstream of the tailrace channel (emission point), the monitoring stations (Figure 1 allowed a longitudinal analysis of the SSCs. The setup of the monitoring program, which considered the fine sediment concentrations along a 160 km-long river section, revealed novel insights concerning the dilution of fine sediment concentrations along the longitudinal profile of the Inn River. SSCs were reduced along the river continuum (as there is usually no additional input in winter), reflecting the hydro-morphological conditions (deposition) as well as dilution due to tributaries. The additionally measured physicochemical parameters at the gauging station Prutz have also never reached critical values [46], which is typical for a high alpine oligotrophic reservoir.

Based on the calibrated continuous suspended sediment data and the discharge, the loads of transported fines during the monitoring period (7 December, 2015–14 March, 2016) was calculated. At the Ried gauging station (reference station) only a small amount of suspended sediment load of approximately 3000 t was transported (Table 2). Due to the discharge of fine material from the Gepatsch reservoir (via the tailrace channel of the HPP Kaunertal), the suspended sediment load at the Inn increased almost tenfold at Prutz during the controlled drawdown. At the measuring points Haiming and Innsbruck, a slightly lower suspended sediment load was calculated than at the measuring station Prutz (Table 2). Overall, in relation to a mean year (2005–2014) at the gauging station Innsbruck, only 1.5% of the annual suspended sediment load was discharged into the Inn River during the drawdown of the Gepatsch reservoir. For comparison, the mean annual load of transported fines at the Innsbruck gauging station is 1.79 million tons. Between the monitoring stations Prutz and Innsbruck, which are approximately 90 km apart, several important tributaries, such as the Sanna River and the River Ötztaler Ache, discharge into the Inn River, which results in an increase in the mean discharge. Data obtained between 2010 and 2014 showed a mean discharge of 78.7 m$^3$ s$^{-1}$ for the Prutz station and 172 m$^3$ s$^{-1}$ for the Innsbruck station. Our data highlights that the input due to the drawdown in winter was relatively high, but far below the natural range of the suspended sediment loads over the year.

**Table 2.** Suspended sediment loads for the time period 07 December, 2015 – 14 March, 2016 at the measuring stations Ried, tailrace channel HPP Kaunertal, Prutz, Haiming, and Innsbruck.

| Measuring Station | Time Period | Suspended Sediment Load (t) Pre-Calibration | Suspended Sediment Load (t) Recalibration |
|---|---|---|---|
| Ried | 07.12.2015–15.03.2016 | +/− 4000 t | +/− 3000 t |
| Tailrace channel HPP Kaunertal | 07.12.2015–15.03.2016 | +/− 26000 t | +/− 23000 t |
| Prutz | 07.12.2015–15.03.2016 | | +/− 27000 t |
| Haiming | 07.12.2015–15.03.2016 | +/− 33000 t | +/−23000 t |
| Innsbruck | 07.12.2015–15.03.2016 | +/− 16000 t | +/− 26000 t |
| Innsbruck | mean year (2005–2014) | 1.79 million tons | n/a |

*3.2. Fine sediment deposits on gravel bars*

The possible retention aspects of fines on the surface layer of the gravel bars were studied on a reach scale presented in this study and on the local scale by obtaining freeze-core and freeze-panel measurements [58]. The results for the first ten gravel bars (Figure 4), from upstream to downstream, showed similar fine sediment distributions at the pre- and post-monitoring stage, but lower deposition heights were measured in March 2016 compared to the measurements conducted in November 2015. For five out of seven gravel bars the means of the fine sediment depositions were higher in March 2016 than in November 2015 (Figure 4).

Moreover, in order to enhance the understanding of the fine sediment depositions, the gravel bars were grouped into three categories expressing their location in the river reach. The first group describes the straight course of the river; the second one includes gravel bars that are situated directly before or after a bend. Additionally, curved sections themselves were analyzed. Conducting the Mann–Whitney U test, no statistically significant changes were observed for any group when comparing the samples

from November with the ones from March. In summary, it must be stated that no significant trend was found in the fine sediment deposition on the gravel bars along the Inn River between Prutz and Wörgl. For 15 out of 34 gravel bars the mean of the fine sediment depositions was lower in March 2016 than in November 2015. Two gravel bars show a statistically significant increase in March 2016 and two, a significant decrease.

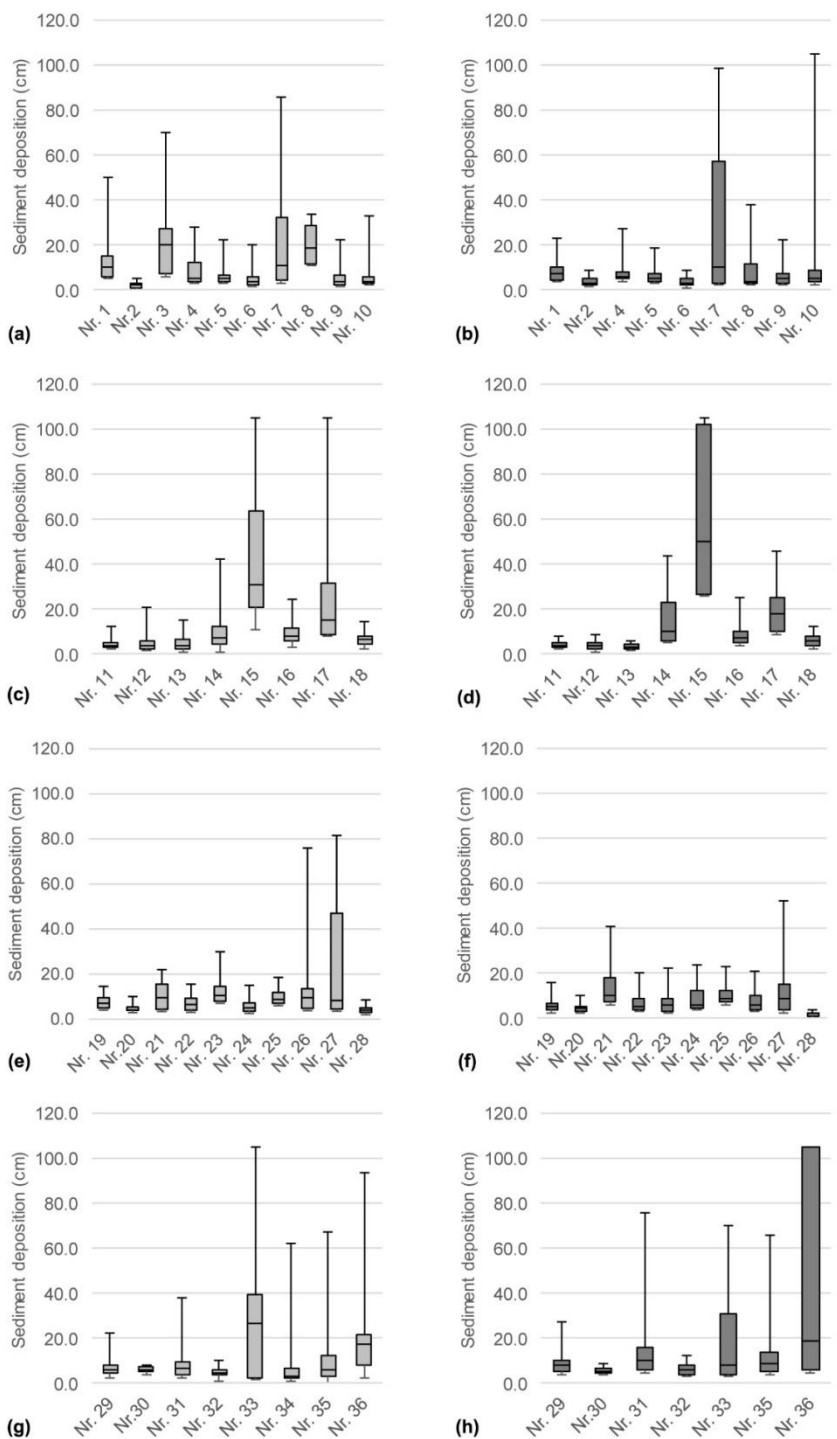

**Figure 4.** Comparison of sediment deposits at selected gravel bars (n = 36) during the pre- (**a,c,e,g**) and post-monitoring (**b,d,f,h**) of the project.

### 3.3. Mapping and Analysis at the Salmonid Spawning Sites

For the evaluation of the multi-stressed river system [73], the results of the mapping potential reproduction sites were important. The results of the fieldwork showed that there was a deficit in the availability of suitable gravel spawning sites along the Inn River. It is noteworthy that 79 potential gravel spawning sites ($d_m$ = 2–5 cm, velocity at the bottom = 0.2–0.4 m s$^{-1}$) were identified at the beginning of the backwater of the Runserau reservoir (located between the gauging station Prutz and the monitoring point and in the Inn residual flow stretch between Runserau and Landeck (Figure 1, Table 3). During the mapping in November 2015, spawning of the brown trout (*Salmo trutta f. fario*) was observed in the Inn residual flow stretch of the HPP Imst (between Runserau and Imst) and documented by means of video recordings. Downstream of the confluence of the Sanna tributary, spawning sites were only identified locally and at specific locations—mostly in the vicinity of gravel-bank structures (restricted dynamic). Due to the general sediment supply deficit in comparison to the (artificially increased) transport capacity of the Inn, extensive spawning habitats covering the entire cross section of the river (for example, upstream riffle areas with gravel $d_m$ = 2–5 cm) do not exist in the reach from Landeck to Innsbruck. Overall, the exiting paved river surface (armored layer of coarse grains on the surface, which acts to protect finer particles underneath from erosion), as result of the reduced bed load supply, but even more impacted due to river regulation and high transport capacity [74], prevents the possible use of spawning grounds, even in those areas which are usable habitat conditions with respect to hydraulics.

**Table 3.** Characteristic grain diameters of gravel spawning probes at the surface and the subsurface layer. SL = surface layer, SSL = subsurface layer, U = uniformity coefficient, and Cc = curvature coefficient.

| Characteristic Grain Diameter (mm) | Fagge Tributary | | Prutz | | Fließ | | Karrösten | | Stams | |
|---|---|---|---|---|---|---|---|---|---|---|
| | SL | SSL | SL | SSL | SL | SSL | SL | SSL | SL | SSL |
| $d_{10}$ | 12.1 | 0.9 | 13.5 | 5.6 | 11.6 | 1.9 | 5.9 | 0.3 | 9.9 | 0.9 |
| $d_{20}$ | 14.4 | 4.5 | 17.3 | 8.8 | 14.1 | 3.5 | 8.8 | 0.7 | 13.7 | 2.0 |
| $d_{30}$ | 16.6 | 8.1 | 20.1 | 11.5 | 17.0 | 5.1 | 10.7 | 3.1 | 16.9 | 2.9 |
| $d_{40}$ | 18.1 | 10.2 | 23.3 | 14.3 | 20.2 | 6.9 | 12.4 | 4.9 | 19.1 | 4.2 |
| $d_{50}$ | 19.7 | 12.3 | 26.4 | 17.5 | 23.5 | 9.1 | 14.1 | 6.6 | 21.7 | 5.2 |
| $d_{60}$ | 21.5 | 14.5 | 30.0 | 21.0 | 26.4 | 11.5 | 16.0 | 8.6 | 24.2 | 6.5 |
| $d_{70}$ | 23.7 | 17.2 | 34.7 | 26.3 | 29.6 | 14.7 | 18.8 | 10.7 | 26.8 | 8.2 |
| $d_{80}$ | 26.5 | 20.5 | 40.7 | 34.0 | 35.1 | 19.5 | 22.1 | 13.5 | 29.7 | 10.7 |
| $d_{90}$ | 29.6 | 25.4 | 47.7 | 45.7 | 44.4 | 27.1 | 30.4 | 17.6 | 37.5 | 15.0 |
| $d_{16}$ | 13.5 | 2.9 | 16.2 | 7.7 | 13.0 | 2.8 | 8.1 | 0.4 | 12.3 | 1.5 |
| $d_{84}$ | 27.7 | 22.0 | 43.4 | 38.3 | 38.6 | 22.1 | 24.9 | 14.7 | 31.0 | 12.1 |
| $d_m$ | **21.0** | **13.2** | **29.1** | **21.9** | **25.6** | **12.4** | **16.9** | **8.1** | **23.1** | **7.2** |
| U | **1.8** | **16.9** | **2.2** | **3.8** | **2.3** | **6.0** | **2.7** | **34.7** | **2.4** | **7.0** |
| Cc | **1.1** | **5.2** | **1.0** | **1.1** | **1.0** | **1.1** | **1.2** | **4.6** | **1.2** | **1.4** |

### 3.4. Fish

Within the electrofishing campaigns 18 fish species were registered, with a dominance of grayling and brown and rainbow trout. Bullhead (*Cottus gobio*) was found at most sites, but not at Fließ. Danube salmon (*Hucho hucho*) were present at Imst, and downstream of Innsbruck. Overall, between Ried and Mils three to six species were found, and this number increased downstream (ten to twelve species). The biomass changed between the different reaches (Figure 5). At the reference site Ried (upstream emission point of suspended sediments) the biomass was between 80 and 90 kg ha$^{-1}$ during both surveys (November 2015 and November 2016). At Prutz (directly downstream of the emission point, with the highest immission) an increase from 65 kg ha$^{-1}$ (November 2015) to 128 kg ha$^{-1}$ (March 2016) was observed, which is related to an increase of the grayling-biomass. The Fließ site is a positive example related to the WFD measure fish lift Runserau, i.e., biomass increased from 16 kg ha$^{-1}$

(November 2015) to 41 kg ha$^{-1}$ (March 2016) as a result of the increased minimum flow (5 m$^3$ s$^{-1}$ instead of 1 m$^3$ s$^{-1}$). At Perjen, in both years, similar shares of brown and rainbow trout as well as bullhead were found, but the grayling was reduced in March 2016 to 10% of its stock from November 2015. This massive decline is likely related to the high SSCs, which were related to the mudflow in the catchment of the Sanna river and subsequent maintenance (September–November 2016). According to our data, this especially affected grayling (mortality and migration), whereas no decline took place for the trout species and the bullhead. This fish-ecological survey revealed that the drawdown of the Gepatsch reservoir had no effects on the fish fauna of the Inn River. Differences between pre- and post-monitoring are due to methodological aspects (e.g., higher biomass at Imst in November 2015, due to the catch of a single Danube salmon) or related to other impacts (i.e., mudflow in the Sanna catchment). Overall, the drawdown did not affect population structure, which at some sites was even better in March 2016 than in November 2015.

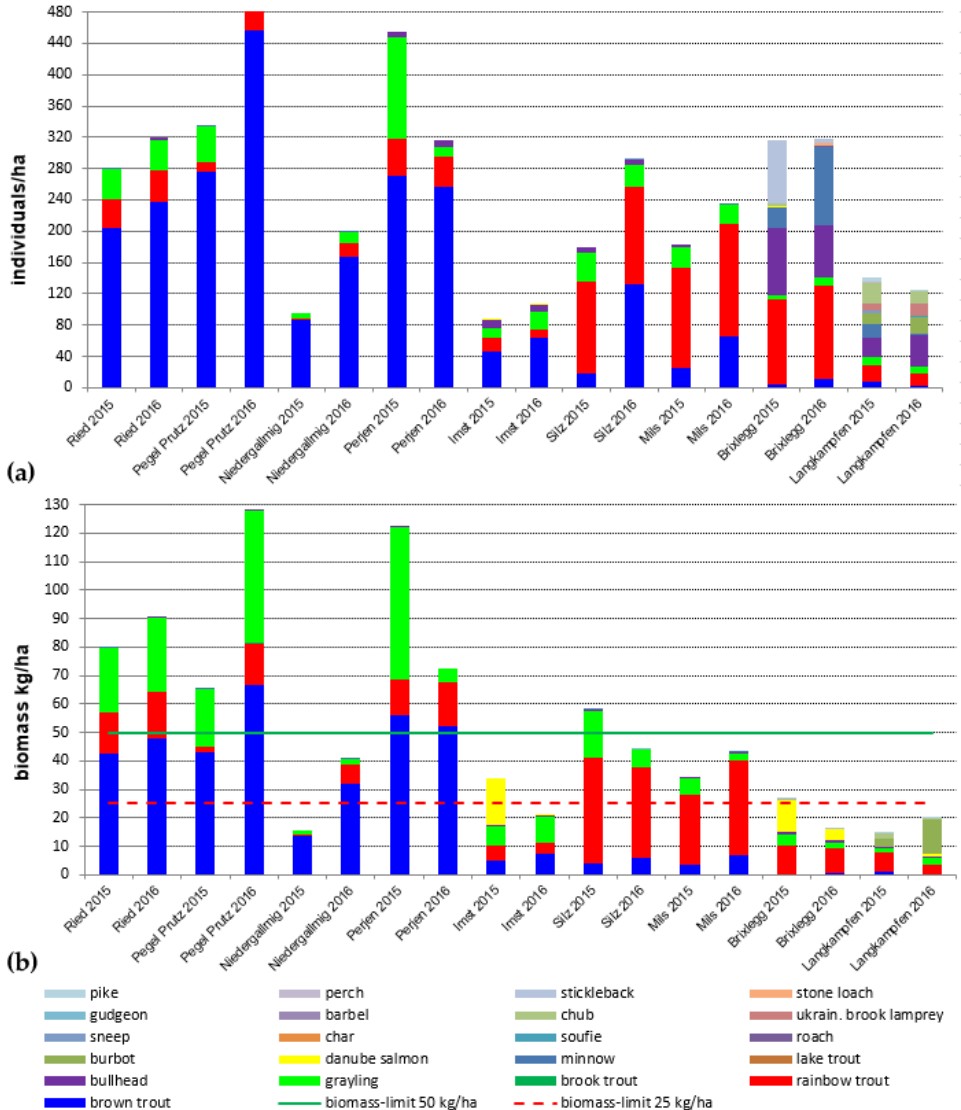

**Figure 5.** Comparison of (**a**) fish abundance and (**b**) biomass for the pre- and post-monitoring. In the "non-glaciated central Alps" fish bioregion (expect sites above 1000 m a.s.l. and rivers with strong sediment transport), a biomass knockout criterion below 50 kg/ha is defined as follows [67]: between 25 and 50 kg/ha corresponds to poor ecological status and locations with biomass values below 25 kg/ha automatically classified as bad ecological status.

### 3.5. Tributary Analysis

For multi-stressed river systems with limited natural reproduction sites in the main channel, the role of tributaries is essential. In our study, out of the 53 total selected tributaries, during the winter low-flow period only 20 tributaries (38%) had sufficient connectivity (according to standard national specifications, e.g., ([75], Annex G,]) for fish. Here, a poorer connectivity is often accompanied with tributaries through steep valley flanks in heavily-regulated sections. With 40% of all tributaries, the most common reason for inadequate connectivity is summarized by the longitudinal interruption of migration routes. In absolute terms, this means that 21 tributaries were due to ramps and 16 tributaries were due to other transversal obstructions with no ascent or fishways for tributary migration. Moreover, the assessment of the upstream sections near the mouth revealed that 51% (27 tributaries) have natural conditions, while the river bed is paved (armored layer as defined in Section 3.3) in 25 tributaries. The additional assessment of the hydro-morphological variability revealed that 62% of the tributaries show more monotonous structures and 34% show more heterogeneous structures. Excluded from both assessment categories are tributaries with piping or enclosures, which account for a total of 4% of the studied creeks. Overall, in 16 tributaries (30%) potential spawning grounds or suitable spawning substrate were detected. However, looking at these places with respect to the backdrop of the fish migration and the connectivity required results in a different picture. Out of the 16 suitable tributaries, only 11 have both spawning gravel and the corresponding connectivity. Proportionately, this means that only 21% of the considered tributaries are in theory suitable for spawning during the winter period. However, hydrological fluctuations in the flow have not been taken into account.

## 4. Discussion

In this section, the single-studied components of the presented reach-scale monitoring are discussed from a holistic perspective.

Fine sediment concentrations in terms of reservoir flushing are one of the drivers for negative impacts in downstream rivers sections [14,22,31,32]. However, the concentrations of suspended sediments caused by a controlled drawdown of the Gepatsch reservoir were significantly higher than the concentrations of suspended sediments normally found in the winter (see reference site Ried). However, the magnitude of concentrations was in a similar order to values of ordinary rainfall events, such as the event in February 2016 (Figure 6a). In this case, SSCs of approximately 1600 mg L$^{-1}$ occurred at the measuring station Landeck. These findings belong to the central aim of the present study, which is to derive recommendations from our comprehensive monitoring for future permits and needs for monitoring (Figure 2). One point of discussion is that thresholds in SSCs are only reflecting possible impacts on instream biota, especially in relation to catchment characteristics in sediment dynamics (e.g., erosion rates, geology, and glaciation). Values that may be extraordinary for Norwegian rivers (sediment production rates <15 t/ha/year; [76]) are only part of the natural lower suspended sediment regime in glacial river systems [77] or rivers with high erosion rates such as Albania (20–40 t/ha/year). This underlines the need for catchment-specific analyses.

This can be even further supported by events in the summer months, exhibited by the natural (partial glacial) hydrology, where very high concentrations of suspended sediments occur in the Inn River. The turbidity at Ried shows average concentrations between 91 and 530 mg L$^{-1}$ during the months of May to September (daily fluctuations due to glacier melting) and varies greatly with extreme values between 16 and 18,302 mg L$^{-1}$ (period 2008–2012) as a result of, e.g., heavy precipitation. Similar monthly mean values (200 to 660 mg L$^{-1}$) are known for the Inn near Innsbruck during the months of May to September [78], as well as a wide fluctuation range for this station. The aquatic biota of the Inn River consequently also has to deal with this strong fluctuation and partly high concentrations. The mean values characterize the glacial regime of the Inn River, which also shapes the aquatic communities and causes extreme living conditions for the aquatic biota. However, during extreme events or through the entry of mudflows into the main river or its tributaries, the SSCs can be significantly exceeded. Mudflows from Dawinbach, Lattenbach, and Mühlbach led to an increase

of the turbidity record at the measuring point Landeck at the Sanna River in September 2016 of over 175,000 mg L$^{-1}$ (Figure 6b). Related to the mudflow event in 2016, at the measuring point Karrösten at least 30,000 mg L$^{-1}$ were recorded by the continuous turbidity measurements (Figure 6b). In comparison, measurements during a 200-year flood event at gauging station Innsbruck showed a value of 25,000 mg L$^{-1}$ of transported sediments as a maximum load. This means that, although the reservoir drawdown caused higher SSCs in the Inn River in terms of the winter low-flow period, the recorded values were orders below the natural high concentration given by the catchment hydrology of the Inn River.

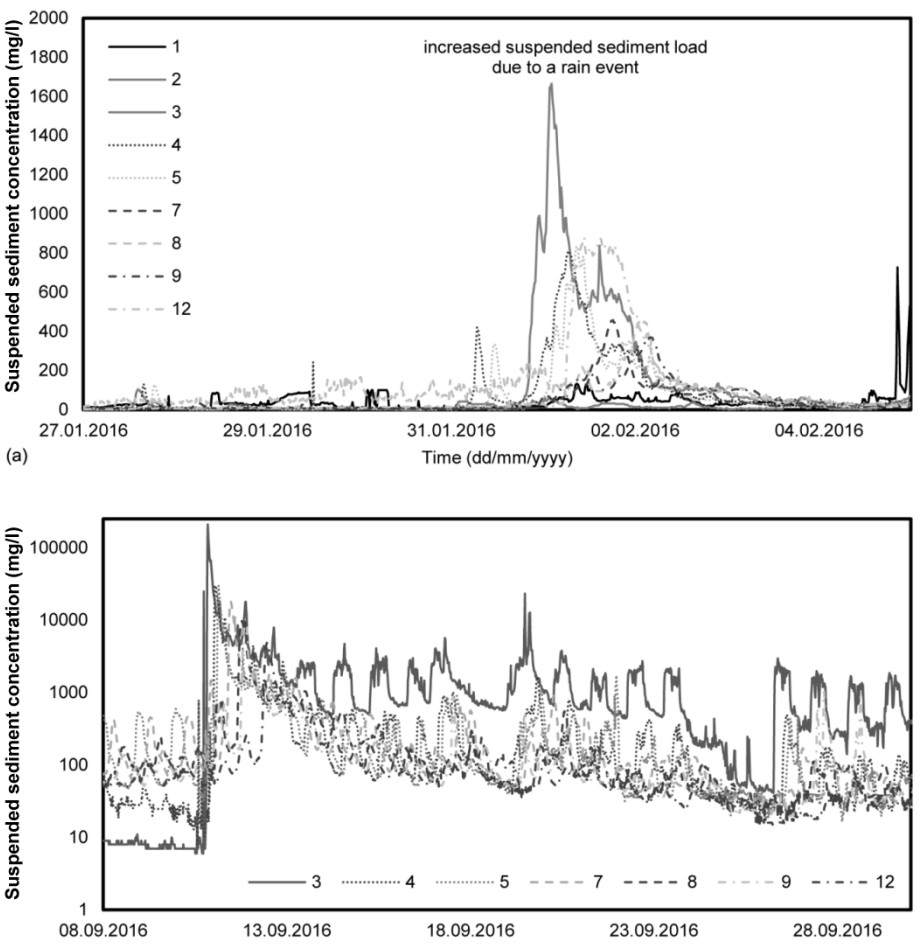

**Figure 6.** (**a**) Natural increase of suspended sediment concentration due to heavy rainfall and (**b**) natural increase in suspended sediment concentration based on a torrent in the tributary catchment of the Sanna River. Daily fluctuations are related to subsequent maintenance works.

Thus, we highlight two important considerations regarding (a) the reliability of point sampling data (like the continuous turbidity sensors) and (b) the feasibility of the existing threshold criteria for harmful impacts in relation to fine sediment concentrations. For these two points of discussion, the habitat-related turbidity measurements delivered important results for dealing with the reliability of the threshold criteria. For instance, [34] classified the effects of increased fine-sediment load as (1) a lethal effect (high-to-low mortality and high-to-medium habitat degradation), (2) lethal and para-lethal effects (high predatory pressure and prolonged hatching of larvae), (3) sub-lethal effects (reduction of growth, fitness and feeding, disturbed homing effect, physiologic stress, and elevated breath frequency), and (4) behavioral effects (emigration and active/passive drift) according to a combination of concentration and duration of increased SSCs. While behavioral effects are mainly reversible and limited to the duration

of exposure, physiological changes can have a more chronic characteristic (review in [79]). Although, studies such as [34] deliver an option for predicting ecological impacts in a first approximation about direct impacts, the results and parts of the discussion (Figure 6) of the presented case study highlight that those thresholds must be interpreted with caution. Additionally, the aspect of suspended sediment concentration variability (Part B; [58]), which was not considered in scientific literature until now, exhibits that due to this variability in turbidity, an option for sheltered habitats, especially for mobile fish might be given. This could be the reason why the use of the Newcombe model [34] would fail when performing an impact assessment for the Inn River based on the single point turbidity measurements in single cross sections.

Another important finding of the study was the detection of a clear lowering in fine sediment concentration due to (i) deposition along the river sensu [80], and (ii) due to an increase in discharge by mixing with tributaries with lower SSCs [81]. Our study underlines the longitudinal decrease of fine sediment concentrations during a controlled drawdown in winter (with low-flow conditions and usually low inputs of SSCs). This has been insufficiently investigated in research projects and publications, but has to be considered in future permits and future monitoring strategies.

In the present study twelve stations for continuous monitoring of SSCs were used. However, this large number would not be required to determine the lowering in concentration along the river Inn. Based on the conducted findings, three to five monitoring stations of continuous recording (including a reference as well as a measuring device at the emission point) should be suitable for future projects, considering the intake of tributaries and distance in the setup of monitoring stations. Similar to hydraulic retention effects, (e.g., [82]) the retention may be non-linear, which should be reflected by the positioning the turbidity loggers downstream of the source of emission.

Gravel bars are important morphological features in the river environment, influencing hydraulics [83,84] and sediment sorting, e.g., [85,86]. Deposition and accumulation of fines on the gravel bar surface is a natural component of river system dynamics, e.g., [87]. In addition, fine sediment deposits on gravel bars are important for riparian-specific plant species (e.g., willows) due to the required moisture for seeds to grow [88,89]. However, the deposits of fines on gravel bars is mostly seen negatively, e.g., [32], especially when they are a result of anthropogenic influences, such as maintenance work at upstream hydropower stations. In the present study, the winter's low-flow conditions in the period of November 2015 to March 2016 did not enable an overtopping of the gravel bars with sediment-laden water, and thus the fine sediment deposits have not changed significantly (the exceptions were two bars with deposition increases and two depositions with decreases). In rivers with an unbalanced equilibrium of bed load supply and transport capacity such as the Inn River, however, the armored surface layer may be prone to severe clogging [71,90]. This must be considered to be a drawback out of the presented data for future studies and permits (Figure 2).

The studies on tributaries of the integrative CDMC downstream of the Gepatsch reservoir (Figure 2) were conducted due to the multi-stressed background of the Inn River, a situation frequently found in European river landscapes, e.g., [91], and due to the specific request for spawning and juvenile habitats; see [92]. The results showed that the proportion of tributaries sufficiently connected to the Inn during the winter's low-flow period is only 38%. The reasons for the lack of connectivity are complex. A significant proportion is attributable to manmade structures, such as sills and ridge ramps, and a smaller part accounts for complete obstructions (piping, enclosures), and only a small number of tributaries are not fish-passable due to natural waterfalls. However, especially in rivers which are multi-stressed, there is a need for sustainable and suitable habitats, e.g., [93,94].

Spawning grounds have been reduced in large and mid-sized rivers in Central Europe to a high extent due to erosive land use and water regulation. In many places this deficiency contributes significantly to the decline of fish stocks. However, the survival of fish is formulated as a political objective and is required by a number of statutory regulations, including the Water Framework Directive and the Natura 2000 network [71]. For this reason, the connectivity of tributaries and suitable spawning grounds is essential to achieve a positive effect through potential successful spawning and

possible growth in the tributaries. This significance for the remediation of severely- impacted water bodies was also referred to as an "ecological anchor" in hydropeaking studies [92], with the background of anchoring the ecological potential of heavily-modified water bodies.

## 5. Conclusions

Based on the case study of the drawdown of the Gepatsch reservoir and the related research questions (what kind of physical and biotic parameters are suitable and needed to detect the impact of increased fine sediment loads during a controlled drawdown on various trophic levels), recommendations are given (Figure 2). Regarding the number of continuously recording SSC monitoring stations, three to five (including a reference as well as a measuring device at the emission) are suitable for future projects. However, a higher number of monitoring points could help to show the effects from local sub-catchments (e.g., tributaries). For an alpine reservoir, it can be concluded that the reach scale monitoring data and especially the reach scale assessment were useful for determining the possible impacts of increased fine sediment loads on the aquatic biota. This, however, could be validated and supported by local scale measurements (Part B; [58]).

Furthermore, it can be concluded that based on the integrative monitoring of the controlled reservoir drawdown in winter, no overtopping of thresholds of the permit was documented and a clear reduction in SSC could be detected along the course of the river Inn. In addition, the suspended sediment retention process is an important issue in sediment management for hydropower stations, which needs to be further investigated in the future. This study revealed insights into an integrative and innovative monitoring concept (technical as well as ecological aspects, the latter including abiotic and biotic parameters) during the controlled release of water from the Gepatsch reservoir. The results showed that according to the 48 h stops during the weekends, even higher concentrations were recorded on Mondays when the operation began again. Thus, an essential conclusion and recommendation for future projects is that continuous releases should be targeted, including periods with reduced discharge, to avoid "artificial" peaks in the SSC in terms of the controlled drawdown of a reservoir. Overall, this study forms a basis for sustainable reservoir management of alpine reservoirs in the future.

**Author Contributions:** conceptualization, C.H. and M.S.; methodology and field work, M.H., P.H., M.H., M.S., and C.H.; investigation: P.F. and H.H.; resources, B.H.; review and editing, C.H., B.W., P.F., and M.S.; visualization, P.F.; funding acquisition, C.H. All authors have read and agreed to the published version of the manuscript.

**Funding:** The fieldwork (monitoring of the controlled drawdown) was carried out on behalf of TIWAG – Tiroler Wasserkraft AG. This paper was written as a contribution to the Christian Doppler Laboratory for Sediment Research and Management. The financial support by the Christian Doppler Research Association, the Austrian Federal Ministry for Digital and Economic Affairs and the Austrian National Foundation for Research, Technology and Development is gratefully acknowledged.

**Conflicts of Interest:** The authors declare no conflict of interest.

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
