# Peer review of "Controlled Reservoir Drawdown—Challenges for Sediment Management and Integrative Monitoring: An Austrian Case Study—Part A: Reach Scale"

_water, doi:10.3390/w12041058_

Round 1

Reviewer 1 Report

Part A of the case study deals with a controlled drawdown of a reservoir Gepatsch (Tyrol, Austria) for the reach scale monitoring data. While the paper has been written well, it is still not clear what the original contribution of the paper is. Maybe it can set up in the Introduction part of the paper with clear objectives. Also, the sample of data used for analysis is too small.

Author Response

Reviewer 1

Part A of the case study deals with a controlled drawdown of a reservoir Gepatsch (Tyrol, Austria) for the reach scale monitoring data. While the paper has been written well, it is still not clear what the original contribution of the paper is. Maybe it can set up in the Introduction part of the paper with clear objectives. Also, the sample of data used for analysis is too small.

Reply: Thanks for the comments and suggestions. In the revised version we gave a focus on highlighting the novelty / original contribution. Especially in the Introduction and Conclusions this novelty was underlined.

Reviewer 2 Report

This paper reports on a detailed field investigation performed during a controlled drawdown of an Alpine hydropower reservoir (Gepatsch Reservoir), located in a tributary valley of the Inn River (Tyrol, Austria). SSC (Suspended Sediment Concentration) and streamflow were monitored for more than 3 months at 12 stations, spanning over 150 km along the Inn River. The field dataset is integrated by before/after assessments of fine-sediment deposition on river bars, as well as of fish assemblages.

In my opinion, the investigated topic, its relevance in contemporary sediment-management in regulated mountain-rivers, the presented dataset, and the related analyses deserve the attention of the international scientific literature.

In the following list, I provide remarks and suggestions to possibly support a further improvement of the manuscript.

INTRODUCTION

Overall, I noticed an interesting similarity between the presented investigation and field investigations carried out in the central Italian Alps and concerning controlled sediment flushing through hydropower reservoirs (recently summarized in PloS one, 14(6), e0218822). This similarity ranges from the investigated setting (i.e., alpine rivers subjected to multiple pressures – L 140-143), to the rationale (including SSC thresholds issued by the concerned authorities) and the methodology. Specifically, the downstream impact of sediment flushing from Cancano Reservoir (i.e., Journal of Environmental Management 182 (2016) 1-12) was studied along a 30 km reach of the Adda River, monitoring both abiotic and biotic variables (see, e.g., your L 81-82, but also 479-484 related to longitudinal decrease of SSC due to dilution/deposition). Storage (120 Mm3), elevation of the dam crest (1,900 m a.s.l.), and electricity generation (i.e., peak energy) are comparable for Cancano and Gepatsch reservoirs. Similarly, sediment flushing through Cancano Reservoir was carried out in winter, i.e., with minimum inflow, and causing concern about severe impact on the early stages of life of brown trout (see, e.g., your L 89-96, but also 114-116).

Of course, as clearly specified in L 29-30, differences between sediment flushing and drawdown are considerable. Moreover, the relevance of the investigated river and the length of the study reach deserve unquestionable attention. However, in my opinion, the mentioned “parallelism” could be highlighted to improve this paper.

Further specific observations

1. “Single field investigation” (L 64) is rather unclear to me, and could be rephrased. If you mean that previous investigations mostly focused on restricted stream reaches, I suggest considering the previously mentioned exceptions.

2. Maximum removal of sediments (L 86) is not necessarily the target of current sediment management by flushing in Alpine reservoirs, particularly when controlling the operation to limit the downstream impact (see previously mentioned PloS one, 14(6), e0218822).

MATERIALS AND METHODS

3. I suggest clearly specifying (L 130) where the target values were required (i.e., according to L 197-198, Prutz).

4. Interruption of the works during the weekends to providing “recovery phases” to the biota (L 135) is another interesting (and still controversial) issue discussed in a mentioned reference (i.e., PloS one, 14(6), e0218822).

5. In order to improve the clarity of the presentation, I suggest providing the names of the sampling stations 1 to 12 (e.g., Ried, Prutz, etc.) in the caption of Figure 1 (L 146-149).

6. The methodology concerning samples collection in the selected spawning grounds (L 236) might be described (see also next point 8).

RESULTS

Overall, I think that basic information concerning SSC and streamflow monitoring along the study period in the mentioned 12 sections could be provided. This could be done in tabular (e.g., average, standard deviation and maximum) and/or in graphical form. In contrast, Figure 3.a provides SSC at Prutz only, and Table 2 is limited to mass-fluxes. Regarding Table 2, additional detail concerning the recalibration procedure might be given in the Methods section. If it concerns a-posteriori calibration of the turbimeters, I observed significant difference between uncalbrated/calibrated fluxes at Haiming/Innsbruck, seeming well above previously reported accuracy (5%, L 172).

Further specific observations

7. In Figure 3.a, during the first phase (L 269-270) SSC at Prutz was frequently in the range 0.5-1 g/l, and occasionally up to 3-4 g/l. Even if complete time-series of SSC are only provided at Prutz, it is reasonable that SSC at the tailrace of Kaunertal HHP was significantly higher. My question is: was the turbine severely damaged? I noticed that specific references are provided (i.e., [45] and [46]), however a brief summary of the effect of such a sediment load on the machinery could be given. Particularly accounting for the very high head of the HPP (around 900 m – L 107).

8. If I understand properly, results of freeze-core sampling are not presented in this paper (L 324-325). See also previous point 6.

9. In L 355 reference to Figure 5 would seem incorrect, did you mean Figure 1?

DISCUSSION

10. SSC time-series reported in Figure 6.b and commented in L 450-452 show very regular daily fluctuations: are you sure that the forcing is limited to natural, though extreme, mudflows?

11. Newcombe & Jensen (1996) classified the effects of increased fine-sediment load rather than considering sediment flushing only (L 464).

12. As already discussed in the literature (see, e.g., the previously mentioned paper PloS one, 14(6), e0218822), the Newcombe & Jensen model can predict impact on fish by increased SSC to a first approximation level. However, the Authors doubts about the model (L 470-478) are not supported by SEV (severity of ill effect) analysis and comparison with observed fish data, not presented in this paper.

Author Response

Reviewer 2

This paper reports on a detailed field investigation performed during a controlled drawdown of an Alpine hydropower reservoir (Gepatsch Reservoir), located in a tributary valley of the Inn River (Tyrol, Austria). SSC (Suspended Sediment Concentration) and streamflow were monitored for more than 3 months at 12 stations, spanning over 150 km along the Inn River. The field dataset is integrated by before/after assessments of fine-sediment deposition on river bars, as well as of fish assemblages.

In my opinion, the investigated topic, its relevance in contemporary sediment-management in regulated mountain-rivers, the presented dataset, and the related analyses deserve the attention of the international scientific literature.

In the following list, I provide remarks and suggestions to possibly support a further improvement of the manuscript.

Reply: Thanks for the positive reply concerning the overall presentation of the research conducted at the controlled reservoir drawdown. Moreover, we are thankful for the suggestions, which were given to improve the quality of the re-submitted MS. In the following all detailed comments given by the reviewer are answered, with some clear descriptions how they were implemented in the revised text.

INTRODUCTION

Overall, I noticed an interesting similarity between the presented investigation and field investigations carried out in the central Italian Alps and concerning controlled sediment flushing through hydropower reservoirs (recently summarized in PloS one, 14(6), e0218822). This similarity ranges from the investigated setting (i.e., alpine rivers subjected to multiple pressures – L 140-143), to the rationale (including SSC thresholds issued by the concerned authorities) and the methodology. Specifically, the downstream impact of sediment flushing from Cancano Reservoir (i.e., Journal of Environmental Management 182 (2016) 1-12) was studied along a 30 km reach of the Adda River, monitoring both abiotic and biotic variables (see, e.g., your L 81-82, but also 479-484 related to longitudinal decrease of SSC due to dilution/deposition). Storage (120 Mm3), elevation of the dam crest (1,900 m a.s.l.), and electricity generation (i.e., peak energy) are comparable for Cancano and Gepatsch reservoirs. Similarly, sediment flushing through Cancano Reservoir was carried out in winter, i.e., with minimum inflow, and causing concern about severe impact on the early stages of life of brown trout (see, e.g., your L 89-96, but also 114-116).

Of course, as clearly specified in L 29-30, differences between sediment flushing and drawdown are considerable. Moreover, the relevance of the investigated river and the length of the study reach deserve unquestionable attention. However, in my opinion, the mentioned “parallelism” could be highlighted to improve this paper.

Reply: Thanks for this important comment and suggestion for improving the quality of the MS. Due to revision work we included additional information of the mentioned projects, which showed similarities to the controlled reservoir drawdown we investigated. The suggested references are helpful, to widen the view on this topic and improving the opportunity to underline the relevance of our study.

Further specific observations

  1. “Single field investigation” (L 64) is rather unclear to me, and could be rephrased. If you mean that previous investigations mostly focused on restricted stream reaches, I suggest considering the previously mentioned exceptions.

Reply: Corrected, like it was suggested by the reviewer. Now, the meaning should be clear to the reader.

  1. Maximum removal of sediments (L 86) is not necessarily the target of current sediment management by flushing in Alpine reservoirs, particularly when controlling the operation to limit the downstream impact (see previously mentioned PloS one, 14(6), e0218822).

Reply: Good point which was considered in the resubmitted version of the article. “Maximum” was deleted in the revised version of the article.

MATERIALS AND METHODS

  1. I suggest clearly specifying (L 130) where the target values were required (i.e., according to L 197-198, Prutz).

Reply: Specified as it was suggested by the reviewer. The “threshold values” were valid for all monitoring stations; however, the point of control, - if the operation had to be stopped – was at Prutz

  1. Interruption of the works during the weekends to providing “recovery phases” to the biota (L 135) is another interesting (and still controversial) issue discussed in a mentioned reference (i.e., PloS one, 14(6), e0218822).

Reply: Thanks for this important comment. We considered this in the resubmitted version of the article.

  1. In order to improve the clarity of the presentation, I suggest providing the names of the sampling stations 1 to 12 (e.g., Ried, Prutz, etc.) in the caption of Figure 1 (L 146-149).

Reply: Done, as it was suggested by the reviewer.

  1. The methodology concerning samples collection in the selected spawning grounds (L 236) might be described (see also next point 8).

Reply: The methodology concerning sample collection in the selected spawning grounds was implemented in the revised version of the article. Thanks for this comment.

RESULTS

Overall, I think that basic information concerning SSC and streamflow monitoring along the study period in the mentioned 12 sections could be provided. This could be done in tabular (e.g., average, standard deviation and maximum) and/or in graphical form. In contrast, Figure 3.a provides SSC at Prutz only, and Table 2 is limited to mass-fluxes. Regarding Table 2, additional detail concerning the recalibration procedure might be given in the Methods section. If it concerns a-posteriori calibration of the turbimeters, I observed significant difference between uncalbrated/calibrated fluxes at Haiming/Innsbruck, seeming well above previously reported accuracy (5%, L 172).

Reply: Important point, thanks for that. We fully considered this important comment in the revised version of the article. We included a new Table concerning average, standard deviation and maximum for the entire monitoring period. Moreover, additional details concerning the recalibration procedure are now implemented in the methods section.

Further specific observations

  1. In Figure 3.a, during the first phase (L 269-270) SSC at Prutz was frequently in the range 0.5-1 g/l, and occasionally up to 3-4 g/l. Even if complete time-series of SSC are only provided at Prutz, it is reasonable that SSC at the tailrace of Kaunertal HHP was significantly higher. My question is: was the turbine severely damaged? I noticed that specific references are provided (i.e., [45] and [46]), however a brief summary of the effect of such a sediment load on the machinery could be given. Particularly accounting for the very high head of the HPP (around 900 m – L 107).

Reply: Another good point, which might be interesting to more people in the research community. However, the turbine abrasion was listed as “local scale” aspect, which needs to be evaluated. Thus, in the revised version of the MS we added some specific reference to part B, were this aspect is addressed in a short (referenced) statement.  

  1. If I understand properly, results of freeze-core sampling are not presented in this paper (L 324-325). See also previous point 6.

Reply: Correct. They are only presented in Part B – local scale aspects of the controlled reservoir drawdown.

  1. In L 355 reference to Figure 5 would seem incorrect, did you mean Figure 1?

Reply: Corrected in the revised MS as it was suggested by the reviewer.

DISCUSSION

  1. SSC time-series reported in Figure 6.b and commented in L 450-452 show very regular daily fluctuations: are you sure that the forcing is limited to natural, though extreme, mudflows?

Reply: This need for clarification was addressed in the resubmitted version of the article.

  1. Newcombe & Jensen (1996) classified the effects of increased fine-sediment load rather than considering sediment flushing only (L 464).

Reply: Yes, you are right. This was corrected / specified in the revised MS.

  1. As already discussed in the literature (see, e.g., the previously mentioned paper PloS one, 14(6), e0218822), the Newcombe & Jensen model can predict impact on fish by increased SSC to a first approximation level. However, the Authors doubts about the model (L 470-478) are not supported by SEV (severity of ill effect) analysis and comparison with observed fish data, not presented in this paper.

Reply. Corrected in the revised article. You are right, that the predicted impacts by the N&J model is just to a first approximation level. Long – term effects are not considered. Moreover, we corrected the non-supported statements concerning the SEV.

Reviewer 3 Report

Dear authors.

The submitted manuscript deals with a very interesting monitoring campaign of a reservoir drawdown which is worth to be published. Part A deals with the reach scale.

Despite the large number of renown researchers, the paper draft in its current form lacks the rigor and scientifically-targeted shape suitable for publication. There are a large number of issues to be addressed prior publication.

The goals of the study are not clear and have to be defined in the introduction. The abstract and introduction are mediocre and do not clarify the novelty of the research project nor the study goals. The discussion is not understandable in terms of both wording and structure.

Further, English lacks clarity and has to be thoroughly improved throughout the entire manuscript except some minor sections (3.4 and 3.5.). A professional proof reading is necessary to obtain the same English level as part B.

Part B is considerably better compared to A in terms of both structure and language. It seems that the authors did not put the same effort in manuscript A, but rather submitted a (very) raw version.

I suggest two different possible ways.

Largely improve manuscript A. Considerably shorten manuscript A and add the remaining text to part B. That is, make one manuscript out of the two.

I tend to way 2, but I leave the final decision to the journal editor, depending also on the other review(s).

Please find further detailed line by line remarks in the following.

Abstract

Rewrite 23 – 25: Sentence too long. Wording.

25: related to the artificial downstream flushing of deposited fines. Rewrite. Ecological impacts are not only due to flushing of fines

27/28 through the possible clogging of the gravel matrix or an increased turbidity and subsequent stress for aquatic species. Unclear, reformulate. What does clogging of gravel matrix mean?

28-30 However, …

Sentence has nothing to do with the text before and is out of context. Rewrite.

Rewrite entire abstract

Introduction

44… hydropower reservoirs. not only hydropower, all reservoirs have this problem.

49: where a reduction of 80% of the potential storage volume by 2035 has been predicted. The same is forecasted for Europe by 2080 [3].

This is vague and needs more precise explanation. How did Basson identify these numbers?

56 -64: These countermeasures are not only for fines but all material. Bypass tunnels are likely to transport mainly the bedload not the fines only.

General remark:

The introduction lacks consistency and a clear structure.

Explain sediment management strategies in a better way. Cite Kondolf et al .2014 (31) and use his three different categories of strategies in order to explain your technique. Explain the difference between sluicing and flushing in the context of your research project. Also line 84-85 does not mention sluicing.

In general, (regular) sluicing diverts the floods (and sediments) directly into the downstream, thereby reestablishing the natural character of a river reach. So, from an ecological point of view, sluicing is the preferred option.

Cite the following references in the following context: These show that not only fines are diverted but also bedload. Recovery is high after short time, hence long term (regular) sediment management is beneficial also for benthic biota.

Auel et al 2017 Effects of sediment bypass tunnels on grain size distribution and benthic habitats in regulated rivers JRBM.

Martín, E.J., Doering, M., and Robinson, C.T., 2017. Ecological assessment of a sediment by-pass tunnel on a receiving stream in Switzerland. River Research and Applications

Serrana et al 2018. Ecological influence of sediment bypass tunnels on macroinvertebrates in dam-fragmented rivers by DNA metabarcoding

Robinson, C.T., Uehlinger, U., and Monaghan, M.T., 2003. Effects of a multi-year experimental flood regime on macroinvertebrates downstream of a reservoir. Aquatic Sciences, 65, 210−222.

Robinson, C.T., Uehlinger, U., and Monaghan, M.T., 2004. Stream ecosystem response to multiple experimental floods from a reservoir. River Research and Applications, 20, 359−377.

89 From an operational point of view, a controlled drawdown is only possible during reduced inflow rates.

Why is that? Explain in more detail

100: State your goals clearly in the end of the introduction. Why are you doing this research, what are the questions to be answered?

Materials and methods

104 ff: indicate the direct and indirect catchment area of gepatsch reservoir.

108: high quality peak load? What does high quality mean in case of energy? Delete and use only peak load.

124: what do you mean by so called free reservoir management applies… is this the name for this one-time flushing event or a general name for a management technique?

146: figure caption. Name all 12 stations in a table or list them in the caption. Indicate Innsbruck and Kufstein in figure b. name the reservoir.

158 figure 2: without further explanation in the text, the flow chart is useless. What is the meaning of the flow chart? delete this figure.

171 explain TS

178 ff cite:

Felix D., Albayrak I., Boes R.M. (2018). In-situ investigation on real-time suspended sedi-ment measurement techniques: Turbidimetry, acoustic attenuation, laser diffraction (LISST) and vibrating tube densimetry. Intl. J. of Sediment Research33: 3–17.

Felix D., Albayrak I., Boes R.M. (2016). Continuous measurement of suspended sediment concentration: Discussion of four techniques. Measurement 89: 44–47.

Show the calibration FNU vs g/l in a figure with all samples taken. Discuss the accuracy of the transformation/calibration.

221: samples of what were pooled, grain size, deposit height?

224 for of. Revise

240: WFD abbreviation for what? Water framework directive

241 Sites of fish survey are not shown in figure 1b. Indicate them in fig 1b.

244 ff, rewrite sentence. Based on the criteria…was calculated before and after the controlled drawdown. Word missing, what was calculated?

248 Table 1. BAW 2010, cite with reference, BMLFUW 2015 cite with reference.

The categories are likely unknown outside of Austria and need to be explained in the text.

256 Land Tirol 2016 – cite as reference

258 n = 180 vs line 251 n = 53 tributaries? So the total was not 53. Rewrite line 251 as e.g.: The Inn tributaries were assessed…

270: penstock etc. What does that mean? Cleaning of the waterway? Describe more precisely. What was cleaned? And why did it take 2 months?

Prutz, specify where that is.

Figure 3: enlarge font. a) intake of what? The plant? Is the intake of the plant not inside the reservoir? what are the grey bars?  Cut figure at 5000 mg/l or use log plot. Put the suspension line in front of the colored bars. Move the letter (a) up. b) log scale is evident, no need to indicate in the caption.

273f: yellow -> is it same to orange (indicated in caption)? Please use the same wording for the caption and the text.

274 ff: Well, high values do not really appear always after the weekend. In the green region (cleaning phase) the concentration is mostly high in the middle of the week.

278ff This paragraph is not correct. You are comparing data from different sites (Innsbruck/Prutz) which does not really allow for a clear statement. At least you should comment on the distance of the two and possible dilution. Maybe values at Prutz are always higher?

Further, you use monthly values in Innsbruck. Explain how the maximum in Innsbruck is calculated? Daily value? 15 min value? The short-term peak might have been considerably higher than your suggested 1000 mg/l. Use the raw data of Innsbruck station.

286: classified short to very short. Is that an official classification in Austria? If not just state the duration without any classification.

296 However, delete. Not correct wording here.

297: emission? Of what? Where? Longitudinal -> how long, some meters, or all along to Germany?

Use more precise wording as e.g. immediately downstream of the flow release of the reservoir…

297-300: sentence has no meaning. Unique insights sound like advertisement but are not an adequate description of your results.

300 turns out that. Revise English. Further, the sentence does not explain anything. Show a figure of the SCC along the Inn. How exactly do you see dilution and how deposition?

this sentence has no context to the one before and after. What do you want to say here?

304ff: This sentence is rather the method and not the result of your analysis.

308: fine material from tailrace: the material is not from the tailrace itself, but from the reservoir. Reformulate.

311: why reference to table 1?

313: State again, how long the drawdown was and make a cross correlation to the annual value. It seems that 1.5% in more than one month drawdown duration is very little.

319 relatively high. Compared to what?

296-320. please rewrite and restructure the entire paragraph. Separate the single topics in own paragraphs.

3.2 Fine sediment deposits

Figure 4: Explain the bars in more detail. What are the boxes showing? Fig 4 has very little meaning for other future research if no correlation is observed. Delete the figure.

In general, this paragraph is OK but has little meaning. Shorten it. For example line 335ff is not needed.

3.3 Mapping and analysis

350F: multi stressed. What are the multi stresses? Why are the results important?

351: deficit before or after drawdown?

354: Where is Runersau reservoir? Never mentioned before

356: what is meant with the residual section of the Inn. Please explain in more detail.

358 severely restricted dynamic structures. How can restricted structures be dynamic? What restricts them? Please explain in more detail.

361: ford

362: Paved: do you mean the armor layer? Maybe better “armored”

365: steady flow velocities. Please explain in more detail. Steady for how long to be adequate for spawning.

367: please prepare a figure instead of a table for this data. Indicate the good area for spawning (2 – 5 cm).

3.4 Fish

Figure 5: name all locations also in fig 1b in order to know where they are. Fig a shows the individuals, b) the biomass. Not pre and post periods. Specify a) and b) in the figure itself

Explain the biomass limits 25/50 kg/ha in the text

384: so, these mudflows are not related to the reservoir drawdown? Please specify

In general, 3.4 is written in good English and comprehensible.

3.5 tributary analysis

399: how is this connectivity defined?

406: please use river bed instead of bottom.

406: Paved: do you mean the armor layer? Maybe better “armored”

412: patency, better use connectivity

In general, 3.5 is written in good English and comprehensible.

Discussion

419: with a differentiation of the scaling instream processes by means of reach scale. Please rewrite and explain. What are scaling instream processes?

However. Delete. Wrong wording

423 of the reservoir Gepatsch. Name it to better understand that you are talking about your monitoring.

428ff: these findings… rewrite sentence. Poor English

430ff: One point of … rewrite sentence. Meaning not clear

433ff What have Norway and Albania to do with the research here? This is out of context. State the values of your study too for comparison.

Fig 6: explain the legend. What are the numbers. Why a and b have different y axis names?

448ff. rewrite. Not clear. What acts as a filter? The glacial regime? Why and how?

453: 200-year flood event.

459: From where do you derive the two discussion points? They have nothing to do with the results discussed before (showing that the SCC by the drawdown are far below the normal summer SCCs).

462: rewrite sentence. Not clear. What are habitat related turbidity measurements? Were they presented in the results?

470ff: Although… rewrite. please write short and clear sentences.

472: How is Fig 5 connected to your discussion? Which thresholds are you talking about?

475: scientific literature. Did you check all worldwide scientific literature to be sure about such a statement? what do you mean by aspect of suspended sediment concentration variability?

480: sensu?

482: same concentrations? As what? Upstream? what has fig 6 to do with your concentrations?

484: name the publications and research projects who are neglecting this aspect.

491 to 497: this is rather the introduction than the discussion.

504 – 514 is a repetition of results. Not needed.

515-523 That paragraph has no connection to the research project.

The entire discussion has to be rewritten and clearly structured. The concept of the discussion remains largely unclear.

525: there was only one question formulated in the introduction.

Rewrite sentence. Not clear. Please use short sentences.

Author Response

Reviewer 3

Formularbeginn

Dear authors.

The submitted manuscript deals with a very interesting monitoring campaign of a reservoir drawdown which is worth to be published. Part A deals with the reach scale.

Despite the large number of renown researchers, the paper draft in its current form lacks the rigor and scientifically-targeted shape suitable for publication. There are a large number of issues to be addressed prior publication.

The goals of the study are not clear and have to be defined in the introduction. The abstract and introduction are mediocre and do not clarify the novelty of the research project nor the study goals. The discussion is not understandable in terms of both wording and structure.

Reply: In the re-submitted version of the article, we improved both the abstract and the introduction to underline the novelty of the article. Moreover, the study goals were once more underlined explicitly.

Further, English lacks clarity and has to be thoroughly improved throughout the entire manuscript except some minor sections (3.4 and 3.5.). A professional proof reading is necessary to obtain the same English level as part B.

Reply: We tried to improve the quality as far as it was possible. However, there is a final professional English proof – reading by the Journal. Thus, all the still given lacking of clarity will be fixed by the Journal.

Part B is considerably better compared to A in terms of both structure and language. It seems that the authors did not put the same effort in manuscript A, but rather submitted a (very) raw version.

I suggest two different possible ways.

Largely improve manuscript A. Considerably shorten manuscript A and add the remaining text to part B. That is, make one manuscript out of the two.

I tend to way 2, but I leave the final decision to the journal editor, depending also on the other review(s).

Reply: Thanks for this suggestion, however, there joining the content of both articles would lead to an overcrowded version B. Thus, we decided to keep the way of presentation the content of the controlled reservoir drawdown in the two articles version.

Please find further detailed line by line remarks in the following.

Reply: Thanks for that, we addressed/answered them point by point.

Abstract

Rewrite 23 – 25: Sentence too long. Wording.

25: related to the artificial downstream flushing of deposited fines. Rewrite. Ecological impacts are not only due to flushing of fines

Reply: Abstract was reworded entirely.

27/28 through the possible clogging of the gravel matrix or an increased turbidity and subsequent stress for aquatic species. Unclear, reformulate. What does clogging of gravel matrix mean?

Reply: Abstract was reworded entirely.

28-30 However, …

Sentence has nothing to do with the text before and is out of context. Rewrite.

Reply: Abstract was reworded entirely.

Rewrite entire abstract

Reply: Corrected in the revised MS as it was suggested by the reviewer

Introduction

44… hydropower reservoirs. not only hydropower, all reservoirs have this problem.

Reply: Specified in the revised MS as it was suggested by the reviewer

49: where a reduction of 80% of the potential storage volume by 2035 has been predicted. The same is forecasted for Europe by 2080 [3].

This is vague and needs more precise explanation. How did Basson identify these numbers?

Reply: We did not change this, as we are not providing a review on Basson`s work…

56 -64: These countermeasures are not only for fines but all material. Bypass tunnels are likely to transport mainly the bedload not the fines only.

Reply:

General remark:

The introduction lacks consistency and a clear structure.

Explain sediment management strategies in a better way. Cite Kondolf et al .2014 (31) and use his three different categories of strategies in order to explain your technique. Explain the difference between sluicing and flushing in the context of your research project. Also line 84-85 does not mention sluicing.

Reply: Change were made in the revised version of the MS as it was suggested by the reviewer.

In general, (regular) sluicing diverts the floods (and sediments) directly into the downstream, thereby reestablishing the natural character of a river reach. So, from an ecological point of view, sluicing is the preferred option.

Reply: Good point. Thanks for this comment. Changes were made as it was suggested by the reviewer.

Cite the following references in the following context: These show that not only fines are diverted but also bedload. Recovery is high after short time, hence long term (regular) sediment management is beneficial also for benthic biota.

Auel et al 2017 Effects of sediment bypass tunnels on grain size distribution and benthic habitats in regulated rivers JRBM.

Martín, E.J., Doering, M., and Robinson, C.T., 2017. Ecological assessment of a sediment by-pass tunnel on a receiving stream in Switzerland. River Research and Applications

Serrana et al 2018. Ecological influence of sediment bypass tunnels on macroinvertebrates in dam-fragmented rivers by DNA metabarcoding

Robinson, C.T., Uehlinger, U., and Monaghan, M.T., 2003. Effects of a multi-year experimental flood regime on macroinvertebrates downstream of a reservoir. Aquatic Sciences, 65, 210−222.

Robinson, C.T., Uehlinger, U., and Monaghan, M.T., 2004. Stream ecosystem response to multiple experimental floods from a reservoir. River Research and Applications, 20, 359−377.

89 From an operational point of view, a controlled drawdown is only possible during reduced inflow rates.

Why is that? Explain in more detail

Reply: In the revised version of the MS, this specific characteristic was explained more in detail as it was suggested by the reviewer.

100: State your goals clearly in the end of the introduction. Why are you doing this research, what are the questions to be answered?

 Reply: Done, as it was already mentioned previously.

Materials and methods

104 ff: indicate the direct and indirect catchment area of gepatsch reservoir.

Reply: Specified in the revised MS as it was suggested by the reviewer. However, not implemented into Figure 1 as scaling of the Figure is not appropriate to highlight such details.

108: high quality peak load? What does high quality mean in case of energy? Delete and use only peak load.

Reply: This is high quality peak load, in terms of flexibility – we included a citation.

124: what do you mean by so called free reservoir management applies… is this the name for this one-time flushing event or a general name for a management technique?

Reply: This is the operation between storage level and minimal operating level, we included an explanation

146: figure caption. Name all 12 stations in a table or list them in the caption. Indicate Innsbruck and Kufstein in figure b. name the reservoir.

Reply: Done like it was suggested by the reviewer.

Reply: We included all stations in the caption and changed the figure

158 figure 2: without further explanation in the text, the flow chart is useless. What is the meaning of the flow chart? delete this figure.

Reply: This figure gives an overview on the project and explains the analyzed parameters (in part A + B)! we therefore want to keep it in!

171 explain TS

Reply: You mean “Solitax ts-line sc from Hach-Lange”? This is the type of sensor, that was used…

178 ff cite:

Felix D., Albayrak I., Boes R.M. (2018). In-situ investigation on real-time suspended sedi-ment measurement techniques: Turbidimetry, acoustic attenuation, laser diffraction (LISST) and vibrating tube densimetry. Intl. J. of Sediment Research33: 3–17.

Felix D., Albayrak I., Boes R.M. (2016). Continuous measurement of suspended sediment concentration: Discussion of four techniques. Measurement 89: 44–47.

Show the calibration FNU vs g/l in a figure with all samples taken. Discuss the accuracy of the transformation/calibration.

Reply: We added this specific request about the calibration Figures as Supplementing material. We don´t think that this needs to presented in the main text.

221: samples of what were pooled, grain size, deposit height?

Reply: we changed “samples” into  measuring points

224 for of. Revise

Reply: Revised accordingly

240: WFD abbreviation for what? Water framework directive

Reply: Revised accordingly

241 Sites of fish survey are not shown in figure 1b. Indicate them in fig 1b.

Reply: It will be difficult to read then; as we included now all station names in the caption of Fig 1b, this is now easy to read and understand with Table 1.

244 ff, rewrite sentence. Based on the criteria…was calculated before and after the controlled drawdown. Word missing, what was calculated?

Reply: Revised accordingly

248 Table 1. BAW 2010, cite with reference, BMLFUW 2015 cite with reference.

The categories are likely unknown outside of Austria and need to be explained in the text.

Reply: Revised accordingly

256 Land Tirol 2016 – cite as reference

Reply: Revised accordingly

258 n = 180 vs line 251 n = 53 tributaries? So the total was not 53. Rewrite line 251 as e.g.: The Inn tributaries were assessed…

Reply: Revised accordingly

270: penstock etc. What does that mean? Cleaning of the waterway? Describe more precisely. What was cleaned? And why did it take 2 months?

Reply: Revised accordingly

Prutz, specify where that is.

Reply: We included a reference to Figure 1b

Figure 3: enlarge font. a) intake of what? The plant? Is the intake of the plant not inside the reservoir? what are the grey bars?  Cut figure at 5000 mg/l or use log plot. Put the suspension line in front of the colored bars. Move the letter (a) up. b) log scale is evident, no need to indicate in the caption.

Reply: The Figure was improved in the resubmitted version of the MS according to the suggestions of the reviewer.

273f: yellow -> is it same to orange (indicated in caption)? Please use the same wording for the caption and the text.

Reply: Revised accordingly

274 ff: Well, high values do not really appear always after the weekend. In the green region (cleaning phase) the concentration is mostly high in the middle of the week.

Reply: The unregularly increases were do to cleaning. Because cleaning was carried out “without water from the reservoir” – in total eight times such artificial “flushing” of the pipe took place

278ff This paragraph is not correct. You are comparing data from different sites (Innsbruck/Prutz) which does not really allow for a clear statement. At least you should comment on the distance of the two and possible dilution. Maybe values at Prutz are always higher?

Reply: Revised accordingly. Well there could be less (“dilution”), however we have similar shares of glaciers also in the catchment area between Prutz and Innsbruck, thus this is a good reference…

Further, you use monthly values in Innsbruck. Explain how the maximum in Innsbruck is calculated? Daily value? 15 min value? The short-term peak might have been considerably higher than your suggested 1000 mg/l. Use the raw data of Innsbruck station.

Reply: Revised accordingly

286: classified short to very short. Is that an official classification in Austria? If not just state the duration without any classification.

Reply: Revised accordingly

296 However, delete. Not correct wording here.

Reply: Revised accordingly

297: emission? Of what? Where? Longitudinal -> how long, some meters, or all along to Germany?

Use more precise wording as e.g. immediately downstream of the flow release of the reservoir…

Reply: Revised accordingly

297-300: sentence has no meaning. Unique insights sound like advertisement but are not an adequate description of your results.

Reply: Revised accordingly

300 turns out that. Revise English. Further, the sentence does not explain anything. Show a figure of the SCC along the Inn. How exactly do you see dilution and how deposition?

Reply: In the resubmitted MS we tried to clarify this specific aspect.

this sentence has no context to the one before and after. What do you want to say here?

Reply: Revised accordingly

304ff: This sentence is rather the method and not the result of your analysis.

Reply: Revised accordingly

308: fine material from tailrace: the material is not from the tailrace itself, but from the reservoir. Reformulate.

Reply: Revised accordingly

311: why reference to table 1?

Reply: Revised accordingly

313: State again, how long the drawdown was and make a cross correlation to the annual value. It seems that 1.5% in more than one month drawdown duration is very little.

Reply: We compare the load during the drawdown with the annual load. This should be clear. Also the timing of the drawdown is clearly shown in 3.1 and Fig 3a; we had 5 days each week, mostly below 1g/l;  the drawdown was stopped on 13.01 due to technical problems. Subsequently a “cleaning” phase startet.

319 relatively high. Compared to what?

Reply: Compared to the reference station Ried (as well as the long term measurements)… à Tab 2

296-320. please rewrite and restructure the entire paragraph. Separate the single topics in own paragraphs.

Reply: Revised accordingly

3.2 Fine sediment deposits

Figure 4: Explain the bars in more detail. What are the boxes showing? Fig 4 has very little meaning for other future research if no correlation is observed. Delete the figure.

Reply: We don´t have the opinion that this Figure is useless in terms of presenting the sampled data on the various gravel bars. The Figure underlines for all sites the high variability of measured fine sediment deposits and that there was almost no deviation between the pre- and postmonitoring. Thus, no changes were made according to this specific point of the reviewer.

In general, this paragraph is OK but has little meaning. Shorten it. For example line 335ff is not needed.

 Reply: Revised accordingly

3.3 Mapping and analysis

350F: multi stressed. What are the multi stresses? Why are the results important?

 Reply: We included a citation

351: deficit before or after drawdown?

Reply: As the mapping took place in in November 2015, it is clear that this is a general deficit à thus tributaries are important for spawning.

354: Where is Runersau reservoir? Never mentioned before

Reply: Revised accordingly – we specified that

356: what is meant with the residual section of the Inn. Please explain in more detail.

Reply: Revised accordingly – we specified that

358 severely restricted dynamic structures. How can restricted structures be dynamic? What restricts them? Please explain in more detail.

Reply: Revised accordingly

361: ford

Reply: Revised accordingly

362: Paved: do you mean the armor layer? Maybe better “armored”

Reply: Revised accordingly

365: steady flow velocities. Please explain in more detail. Steady for how long to be adequate for spawning.

Reply: Revised accordingly

367: please prepare a figure instead of a table for this data. Indicate the good area for spawning (2 – 5 cm).

Reply: No changes were made according to this suggestion in the revised version of the MS. However, we added a Figure of the grain size distributions as supplementing material.

3.4 Fish

Figure 5: name all locations also in fig 1b in order to know where they are. Fig a shows the individuals, b) the biomass. Not pre and post periods. Specify a) and b) in the figure itself

Reply: Revised accordingly

Explain the biomass limits 25/50 kg/ha in the text

Reply: We included this information accordingly.

384: so, these mudflows are not related to the reservoir drawdown? Please specify

Reply: Correct, they are not related to the drawdown, as this happened in another catchment. Also it is outside the time period of the permit and of course it is clear wherefrom the SSCs came (as there were many monitoring sites [still installed])…

In general, 3.4 is written in good English and comprehensible.

3.5 tributary analysis

399: how is this connectivity defined?

Reply: Connectivity was determined according the national (Austrian) standards for the implementation of the EU Water Framework Directive. Requested citation according to this standard was implemented in the resubmitted version.

406: please use river bed instead of bottom.

Reply: Revised accordingly

406: Paved: do you mean the armor layer? Maybe better “armored”

Reply: We included a specification

412: patency, better use connectivity

Reply: Revised accordingly

In general, 3.5 is written in good English and comprehensible.

Discussion

419: with a differentiation of the scaling instream processes by means of reach scale. Please rewrite and explain. What are scaling instream processes?

However. Delete. Wrong wording

Reply: Revised accordingly

423 of the reservoir Gepatsch. Name it to better understand that you are talking about your monitoring.

Reply: Revised accordingly

428ff: these findings… rewrite sentence. Poor English

Reply: Revised accordingly

430ff: One point of … rewrite sentence. Meaning not clear

Reply: Revised accordingly

433ff What have Norway and Albania to do with the research here? This is out of context. State the values of your study too for comparison.

Reply: We specified that. We did not analyze sediment production rates in our study, this is just to exemplify the range we have in Europe…

Fig 6: explain the legend. What are the numbers. Why a and b have different y axis names?

448ff. rewrite. Not clear. What acts as a filter? The glacial regime? Why and how?

Reply: We specified that.

453: 200-year flood event.

Reply: Revised accordingly

459: From where do you derive the two discussion points? They have nothing to do with the results discussed before (showing that the SCC by the drawdown are far below the normal summer SCCs).

Reply: We specified that.

462: rewrite sentence. Not clear. What are habitat related turbidity measurements? Were they presented in the results?

Reply: We specified that, this is a link to part B.

470ff: Although… rewrite. please write short and clear sentences.

Reply: Although, the sentence was reworded, it is still a longer one compared to other parts of the MS. However, we think that this new version is fine.

472: How is Fig 5 connected to your discussion? Which thresholds are you talking about?

Reply: We corrected that, here we refer to Fig. 06.

475: scientific literature. Did you check all worldwide scientific literature to be sure about such a statement? what do you mean by aspect of suspended sediment concentration variability?

Reply: We have a good overview on the literature and currently work as a leading group (Christian Doppler Laboratory for Sediment Research and Management) on this topic. This was studied in part B -- > a reference is included; but at the same time this “local aspect” is a great importance for mobile fish (reach scale), this is why we discuss this here.

480: sensu?

Reply: We want to highlight, that we used it "in the sense of" (= sensu)

482: same concentrations? As what? Upstream? what has fig 6 to do with your concentrations?

Reply: We specified that

484: name the publications and research projects who are neglecting this aspect.

Reply: As reviewer two had some concerns about the general statement given in the initial version of the MS; - This part was reworded in terms of revision and thus no additional references are requested at that point of the text.

491 to 497: this is rather the introduction than the discussion.

Reply: Unfortunately, this statement is somehow unclear to us. Some basics needs to be picked up in the discussion chapter, to (i) interpret the own results and (ii) allow the reader to follow the main outline of the article.  

504 – 514 is a repetition of results. Not needed.

Reply: Similar to the above given reply. We think some minor repetition of results is needed for an interpretation / discussion of the own research. Not only referencing to the results chapter.

515-523 That paragraph has no connection to the research project.

Reply: The meaning of that comment was not clear for us. Thus, no specific changes according to this comment were done.

The entire discussion has to be rewritten and clearly structured. The concept of the discussion remains largely unclear.

525: there was only one question formulated in the introduction.

Rewrite sentence. Not clear. Please use short sentences.

Reply: We specified that

Round 2

Reviewer 3 Report

Dear authors

Congratulations and thank you very much for the very sound and detailed review of the original MS. Many comments were directly incorporated and the MS itself reads well.

The MS improved considerably and is now in line with the high quality of part B.